# MODEL ZOO: A GROWING "BRAIN" THAT LEARNS CONTINUALLY

**Rahul Ramesh & Pratik Chaudhari**
University of Pennsylvania
{rahulram,pratikac}@seas.upenn.edu

## ABSTRACT

This paper argues that continual learning methods can benefit by splitting the capacity of the learner across multiple models. We use statistical learning theory and experimental analysis to show how multiple tasks can interact with each other in a non-trivial fashion when a single model is trained on them. The generalization error on a particular task can improve when it is trained with synergistic tasks, but can also deteriorate when trained with competing tasks. This theory motivates our method named Model Zoo which, inspired from the boosting literature, grows an ensemble of small models, each of which is trained during one episode of continual learning. We demonstrate that Model Zoo obtains large gains in accuracy on a wide variety of continual learning benchmark problems.

## 1 INTRODUCTION

A continual learner seeks to leverage data from past tasks to learn new tasks shown to it in the future, and in turn, leverage data from these new tasks to improve its accuracy on past tasks. It stands to reason that the performance of such a learner would depend upon the relatedness of these tasks. If the two sets of tasks are dissimilar, learning on past tasks is unlikely to benefit future tasks—it may even be detrimental. And similarly, new tasks may cause the learner to "forget" and result in deterioration of accuracy on past tasks. Our goal in this paper is to model the relatedness between tasks and develop new methods for continual learning that result in good forward-backward transfer by accounting for similarities and dissimilarities between tasks. Our contributions are as follows.

**1. Theoretical analysis.** We characterize when multiple tasks can be learned using a single model and, likewise, when doing so is detrimental to the accuracy of a particular task. The key technical idea here is to define a notion of relatedness between tasks. We first show how if the inputs of different tasks are "simple" transformations of each other (and likewise for the outputs), then one can learn a shared feature generator that generalizes better on every task, compared to training that task in isolation. Such tasks are strongly related to each other and therefore it is beneficial to fit a single model on all of them. We show that if tasks are not so strongly related, in particular if the optimal model for one task predicts poorly on another task, then fitting a single model on such tasks may be worse than training each task in isolation. Such tasks *compete* with each other for the fixed capacity in the single model. We also empirically study this competition using the CIFAR-100 dataset.

**2. Algorithm development.** The above analysis suggests that a continual learner could benefit from splitting its learning capacity across sets of synergistic tasks. We develop such a continual learner called Model Zoo. At each episode, a small multi-task model that is fitted to the current task and some of the past tasks is added to Model Zoo. This method is loosely inspired from AdaBoost in that it selects tasks that performed poorly in the past rounds and could therefore benefit the most from being trained with the current task. At inference time, given the task, we average predictions from all models in the ensemble that were trained on that task.

**3. Empirical results.** We comprehensively evaluate Model Zoo on existing task-incremental continual learning benchmark problems and show comparisons with existing methods. There is a wide variety in the problem settings used by existing methods, e.g., some replay data from past tasks (like Model Zoo is designed to do), some replay only a subset of data, some train only for one epoch in each episode, some use extremely small architectures, etc. We compare Model Zoo with existing methods in a number of these settings. **Model Zoo obtains better accuracy than existing methods on the evaluated benchmarks**. Improvement in average per-task accuracy is quite large in some cases, e.g., 30% for Split-miniImagenet. We also show that Model Zoo demonstrates strong forward and backward transfer.

**4. A critical look at continual learning.** We find that even an Isolated learner, i.e., one which trains a (small) model on tasks from each episode and does not perform any continual learning, significantly

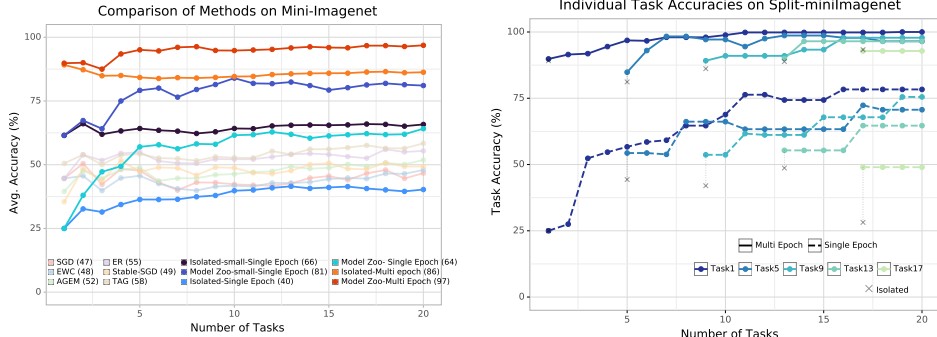

**Figure 1: Left: How well do existing continual learning methods work?** We track the average accuracy (over all tasks seen until the current episode) on the Split-miniImagenet dataset and compare our method Model Zoo and its variants (all in bold) to existing continual learning methods (faint lines, see Table 1 for references). All methods in this plot (unless specified otherwise) use the single epoch setting, i.e., each new task is allowed only 1 epoch of training. Isolated refers to a very simplistic realization of Model Zoo where a separate model is fitted at each episode without any continual learning, or data sharing between tasks; Isolated-small or Model Zoo-small refer to using a very small deep network with 0.12M weights. A number of surprising findings are seen here. (i) Isolated-small (black) outperforms existing methods by more than 10% margin, while having a faster training time, inference time, comparable model size and without performing any data replay. This indicates that **existing methods do not sufficiently leverage data from multiple tasks**. This also indicates the utility of simple methods like Isolated to perform a more prosaic, matter-of-fact, evaluation of continual learning. (ii) While the larger model with 3.6M weights per round, Isolated-Single Epoch (royal blue), performs poorly, its accuracy is better than all existing methods (Isolated-Multi Epoch) upon being trained for multiple epochs. This indicates that **existing methods may be severely under-trained in the single-epoch setting** and this may not be the appropriate setting to build continual learning methods; this was also noticed by Lopez-Paz and Ranzato (2017). (iii) Model Zoo and Model Zoo-small which replay all data from past tasks (A-GEM also replays 10% of the data), achieves around 10% improvement over its Isolated counterparts in both the single-epoch and multi-epoch setting; all these 4 **methods advocated in this paper are better than existing algorithms**. Even Model Zoo-single epoch which replays past data but trains on the new task only for 1 epoch outperforms existing methods significantly. This indicates that replaying data from past tasks is beneficial (Robins, 1995), even if replay may not conform to certain stylistic formulations of continual learning in the literature (Farquhar and Gal, 2019a; Kaushik et al., 2021). Not doing so significantly hurts forward and backward transfer, and average task accuracy.

**Right: Does the single-epoch setting show forward-backward transfer?** The evolution of individual task accuracy of Model Zoo (the multi-epoch setting in bold and single-epoch setting in dotted), on the Split-miniImagenet dataset (only 5 tasks are plotted here, see Fig. A6 for the full version). The X markers denote the accuracy of Isolated. Accuracy of tasks improves with each episode which indicates backward transfer. Also, the X markers are often below the initial accuracy of the task during continual learning, which indicates forward transfer. While both single-epoch and multi-epoch Model Zoo show good forward-backward transfer, the accuracy of tasks for the former is about 25% worse than the latter; corresponding plots for other methods are in Appendix B.7. This indicates that we should also pay attention to under-training and per-task accuracy in continual learning.

outperforms *most* existing continual learning methods on the evaluated benchmark problems, e.g., by more than 8% in Fig. 1 and Tables 1 and A1. This strong performance is surprising because it is a very simple learner that has better training/inference time, no data replay, and a comparable number of weights as that of existing methods.

## 2 A THEORETICAL ANALYSIS OF HOW TO LEARN FROM MULTIPLE TASKS

In this section, we (i) formulate the problem of learning from multiple tasks, (ii) discuss a simple model that highlights when training one model on multiple tasks is beneficial, and (iii) show new results on how the fixed capacity of the model causes competition between tasks.

### 2.1 PROBLEM FORMULATION

A supervised learning task is defined as a joint probability distribution $P(x, y)$ of inputs $x \in X$ and labels $y \in Y$. The learner has access to $m$ i.i.d samples $S = \{x_i, y_i\}_{i=1,...,m}$ from the task. A hypothesis is a function $h : X \to Y$ with $h \in H$ being the hypothesis space. The learner may select a hypothesis that minimizes the empirical risk $\hat{e}_S(h) = \frac{1}{m} \sum_{i=1}^{m} \mathbf{1}_{\{h(x_i) \neq y_i\}}$ with the hope of achieving a small population risk $e_P(h) = \mathbb{P}(h(x) \neq y)$. Classical PAC-learning results (Vapnik, 1998) suggest that with probability at least $1 - \delta$ over draws of the data $S$, uniformly for any $h \in H$, we have $e_P(h) \leq \hat{e}_S(h) + \epsilon$ if

$$m = \mathcal{O}\big((D - \log \delta) / \epsilon^2\big) \tag{1}$$

where $D = \text{VC}(H)$ is the VC-dimension of the hypothesis space $H$. We define the "excess risk" of a hypothesis as $\mathcal{E}_P(h) = e_P(h) - \inf_{h \in H} e_P(h)$. In the continual learning setting, a new task is shown to the learner at each episode (or round). Hence after $n$ episodes, the learner is presented with $n$ tasks $\bar{P} := (P_1, \ldots, P_n)$, with the corresponding training sets $\bar{S} := (S_1, \ldots, S_n)$, each with $m$ samples, and the learner selects $n$ hypotheses $\bar{h} = (h_1, \ldots, h_n) \in H^n$, each $h_i \in H$. If it seeks a small average population risk $e_{\bar{P}}(\bar{h}) = \frac{1}{n}\sum_{i=1}^{n} e_{P_i}(h_i)$, it may do so by minimizing the average empirical risk $\hat{e}_{\bar{S}}(\bar{h}) = \frac{1}{n}\sum_{i=1}^{n} \hat{e}_{S_i}(h_i)$. As Baxter (2000) shows, under very general conditions, if

$$m = \mathcal{O}\big(\epsilon^{-2}\left(d_H(n) - 1/n \log \delta\right)\big), \tag{2}$$

then we have $e_{\bar{P}}(\bar{h}) \leq e_{\bar{S}}(\bar{h}) + \epsilon$ for any $\bar{h} \in H^n$. The quantity $d_H(n)$ here is a generalized VC-dimension for the family of hypothesis spaces $H^n$, which depends on the joint distribution of tasks. Larger the number of tasks $n$, smaller the $d_H(n)$ (Ben-David and Borbely, 2008). Whether (2) is an improvement upon training the task in isolation as in (1) depends upon the hypothesis class $H$ and the relatedness of tasks $P_1, \ldots, P_n$ through the quantity $d_H(n)$. The most important thing to note here is that according to these calculations, if one wishes to obtain a small *average* population risk across tasks, training multiple tasks together cannot be worse: $d_H(n) \leq \text{VC}(H)$.

## 2.2 Controlling the excess risk of a specific task for synergistic tasks

An important goal of continual learning is to have low risk on *all tasks*. This is a stronger requirement than for (2) which bounds the *average* population risk on all tasks.

Suppose there exists a family $F$ of functions $f_i : X \to X$ that map the inputs of one task to those of another, i.e., any task can be written as

$$P_j(A) = f[P_i](A) = \mathbb{P}_i(\{(f(x), y) : (x, y) \in A\})$$

for some function $f \in F$ for any set $A$. We can assume without loss of generality that $F$ acts as a group over the hypothesis space and $H$ is closed under its action. In simple words, this entails that given $h \in H$ suitable for task $P$, we can obtain a new hypothesis $h \circ f$ that is suitable for another task $f[P]$. Instead of searching over the entire space $H^n$ like in §2.1, we now only need to find a hypothesis $h \in H$ such that its orbit

$$[h]_F = \{h' : \exists f \in F \text{ with } h' = h \circ f\}$$

contains hypotheses that have low empirical risk on each of the $n$ tasks. Conceptually, this step learns the inductive bias (Baxter, 2000; Thrun and Pratt, 2012). The sample complexity of doing so is exactly (2). From within this orbit, we can select a hypothesis that has low empirical risk for a chosen task $P_1$. The sample complexity of this second step is

$$|S_1| = \mathcal{O}\big(\epsilon^{-2}\left(d_{\max} - \log \delta\right)\big) \tag{3}$$

where $d_{\max} = \sup_{h \in H} \text{VC}([h]_F)$. By uniform convergence, as Ben-David and Schuller (2003) show, this two-step procedure assures low excess risk for *every* task $P_1, \ldots, P_n$. We have

$$\sup_{h \in H} \text{VC}([h]_F) = d_{\max} \leq d_H(n+1) \leq d_H(n) \leq D = \text{VC}(H). \tag{4}$$

The total sample complexity is favorable to that of learning the task in isolation if both $d_H(n)$ and $d_{\max}$ are small. For instance, if $F$ is finite and $n/\log n \geq D$, we have $d_H(n) \leq 2 \log |F|$ which indicates that we get a statistical benefit of learning with multiple tasks if $D \gg \log |F|$.

**Remark 1 (Data from other tasks may not improve accuracy even if they are synergistic).** Let us make a few observations using the above analysis. (i) From (4), number of samples per task $m$ decreases with $n$; this is the benefit of the strong relatedness among tasks and as we see next, this is *not* the case in general. (ii) The number of tasks scales essentially linearly with $D$, which indicates that one should use a small model if we have few tasks. (iii) But we cannot always use a small model. If tasks are diverse and related by complex transformations with a large $|F|$, we need a large hypothesis space to learn them together. If $|F|$ is large and $H$ is not appropriately so, the VC-dimension $d_{\max}$ is as large as $D$ itself; in this case there is *again no statistical benefit* of training with multiple tasks together, but there is no deterioration either.

## 2.3 Task competition occurs for hypothesis spaces with limited capacity

There could be settings under which fitting one model on multiple tasks may not suffice. To study this, we consider a weaker notion of relatedness. We say that two tasks $P_i, P_j$ are $\rho_{ij}$-related if

$$c \, \mathcal{E}_{P_i}^{1/\rho_{ij}}(h) \geq \mathcal{E}_{P_j}(h, h_i^*), \text{ for all } h \in H. \tag{5}$$

Here $\mathcal{E}_P(h, h') := e_P(h) - e_P(h')$ and $h_i^* = \operatorname{argmin}_{h \in H} e_{P_i}(h)$ is the best hypothesis for task $P_i$; we set $c \geq 1$ to be a coefficient independent of $i, j$. Smaller the $\rho_{ij}$, more useful the samples from $P_i$ to learn $P_j$. The definition suggests that all hypotheses $h$ which have low excess risk on $P_i$ also have low excess risk on $P_j$ up to an additive term $e_{P_j}(h^*)$ and this effect becomes stronger as $\rho_{ij} \to 1_+$. Note that the definition of relatedness is not symmetric. Hanneke and Kpotufe (2020) call

this the transfer exponent. To gain some intuition, we can connect this definition to a certain triangle inequality between the tasks developed by Crammer et al. (2008): in the realizable setting where $e_{P_i}(h_i^*) = 0$, for $c, \rho_{ij} = 1$, we can write (5) as

$$e_{P_i}(h) + e_{P_j}(h_i^*) \geq e_{P_j}(h)$$

which is akin to a triangle with vertices at $h, h_i^*$ and $h_j^*$ with terms like $e_{P_i}(h)$ representing the length of the side between $h$ and $h_i^*$. This definition therefore models a set of tasks and hypothesis space that is not unduly pathological, $e_{P_j}(h)$ cannot be much worse than the sum of the other two sides. We can now show the following theorem bounds the excess risk $\mathcal{E}_{P_1}(h)$ for a hypothesis $h$ trained using data from multiple tasks. See Appendix C for the proof.

**Theorem 2 (Task competition).** Say we wish to find a good hypothesis for task $P_1$ and have access to $n$ tasks $P_1, \ldots, P_n$ where each pair $P_i, P_j$ are $\rho_{ij}$-related. Arrange tasks in an increasing order of $\rho_{i1}$, i.e., their relatedness to $P_1$. Let this ordering be $P_{(1)}, P_{(2)}, \ldots, P_{(n)}$ with $\rho_{(1)} \leq \rho_{(2)} \leq \cdots \leq \rho_{(n)}$ and $P_{(1)} \equiv P_1$ and $\rho_{(1)} = 1$. Let $\hat{h}^k$ be the hypothesis that minimizes the average empirical risk of the first $k \leq n$ tasks. Then, with probability at least $1 - \delta$ over draws of the training data,

$$\mathcal{E}_{P_1}(\hat{h}^k) \leq \frac{1}{k}\sum_{i=1}^{k} \mathcal{E}_{P_1}(h_{(i)}^*) + \frac{c}{k}\left(e_{\bar{S}}(h) + c'\left(\frac{D - \log\delta}{km}\right)^{1/2}\right)^{1/\rho_{\max}} \tag{6}$$

where $\rho_{\max}(k) = \max\left\{\rho_{(1)}, \ldots, \rho_{(k)}\right\}$ and $c, c'$ are constants.

Notice that the first term grows with the number of tasks $k$ because we pick tasks with lower $\rho_{i1}$ that are more and more dissimilar to $P_1$. The second term typically decreases with $k$. The empirical risk $e_{\bar{S}}(h)$ is typically small; in our experiments with deep networks we achieve essentially zero training error on all. Increasing the number of tasks $k$, increases the effective number of samples $km$, thereby reducing the second term in totality. At the same time, these new samples are increasingly more inefficient because $\rho_{\max}(k)$ increases with $k$.

**Remark 3 (Picking the size of the hypothesis space).** The first and second terms characterize synergies and competition between tasks and balancing them is the key to good performance on a given task. Increasing the size of the hypothesis space reduces the first term since it allows a single hypothesis to more easily agree on two distinct distributions $P_i$ and $P_j$. However, this comes at the cost of increasing the second term which grows with the size of the hypothesis space.

**Remark 4 (The set of synergistic tasks can be different for different tasks).** The right hand side in (6) is minimized for a choice of $k$ (where $1 \leq k \leq n$) that balances the first and second terms. The optimal $k$ can vary with the task, e.g., for generic tasks most other tasks will be synergistic and similarly a small optimal $k$ indicates task dissonance where the particular task, say $P_1$ should be trained on with a specific set of other tasks. Even for typical datasets like CIFAR-100, it is highly nontrivial to understand the ideal set of tasks to train with; Fig. 2 studies this experimentally.

**Remark 5 (Continual learning is particularly challenging due to task competition).** Theorem 2 indicates that not only is the learner shown tasks sequentially, but it also may have to work against the competition between the current task and the representation learned on a past task. It does not have access to synergistic tasks from the future while learning on the current task. And further, in settings where there is no data replay, the learner cannot benefit from past synergistic tasks explicitly, other than the representation that it has already learnt. This suggests that one must be even more careful about how the representation in continual learning should be updated.

## 3 MODEL ZOO: A CONTINUAL LEARNER THAT GROWS ITS LEARNING CAPACITY

Theorem 2 can be thought of as a "no free lunch theorem". It indicates that ones should not always expect improved excess risk by combining data from different tasks. This theorem also suggests a way to work around the problem via Remarks 3 and 4. If we learn small models on synergistic tasks, we can hope to have each task benefit from the synergies without deterioration of accuracy due to task competition with dissonant tasks. Model Zoo is a simple method that is designed for this purpose.

Let us assume that tasks $P_1, \ldots, P_n$ are shown sequentially to the continual learner. We assume that all tasks have the same input domain $X$ but may have different output domains $Y_1, \ldots, Y_n$. At each "episode" $k$, Model Zoo is designed to train using the current task $P_k$ and a subset of the past tasks. For example, at episode $k = 2$, we train a model with a feature generator $h$ and task-specific classifiers to obtain models $g_1 \circ h : X \mapsto Y_1$ and $g_2 \circ h : X \mapsto Y_2$. This model can classify inputs from both tasks and gives out a probability vector $p_{g_i \circ h}(y \mid x), \forall y \in Y_i$ depending upon the task. We assume that the identity of the task is known at the test time.

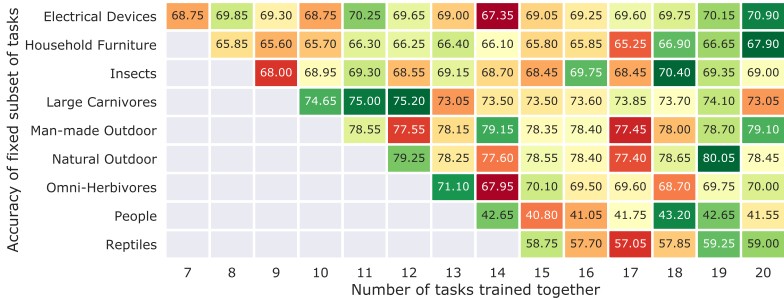

| | 7 | 8 | 9 | 10 | 11 | 12 | 13 | 14 | 15 | 16 | 17 | 18 | 19 | 20 |
|---|---|---|---|---|---|---|---|---|---|---|---|---|---|---|
| Electrical Devices | 68.75 | 69.85 | 69.30 | 68.75 | 70.25 | 69.65 | 69.00 | 67.35 | 69.05 | 69.25 | 69.60 | 69.75 | 70.15 | 70.90 |
| Household Furniture | | 65.85 | 65.60 | 65.70 | 66.30 | 66.25 | 66.40 | 66.10 | 65.80 | 65.85 | 65.25 | 66.90 | 66.65 | 67.90 |
| Insects | | | 68.00 | 68.95 | 69.30 | 68.55 | 69.15 | 68.70 | 68.45 | 69.75 | 68.45 | 70.40 | 69.35 | 69.00 |
| Large Carnivores | | | | 74.65 | 75.00 | 75.20 | 73.05 | 73.50 | 73.50 | 73.60 | 73.85 | 73.70 | 74.10 | 73.05 |
| Man-made Outdoor | | | | | 78.55 | 77.55 | 78.15 | 79.15 | 78.35 | 78.40 | 77.45 | 78.00 | 78.70 | 79.10 |
| Natural Outdoor | | | | | | 79.25 | 78.25 | 77.60 | 78.55 | 78.40 | 77.40 | 78.65 | 80.05 | 78.45 |
| Omni-Herbivores | | | | | | | 71.10 | 67.95 | 70.10 | 69.50 | 69.60 | 68.70 | 69.75 | 70.00 |
| People | | | | | | | | 42.65 | 40.80 | 41.05 | 41.75 | 43.20 | 42.65 | 41.55 |
| Reptiles | | | | | | | | | 58.75 | 57.70 | 57.05 | 57.85 | 59.25 | 59.00 |

Accuracy of fixed subset of tasks (y-axis) / Number of tasks trained together (x-axis)

**Figure 2: Competition between tasks in continual learning can be non-trivial.** In order to demonstrate how some tasks help and some tasks hurt each other, we run a multi-task learner for a varying number of tasks (X-axis) and track the accuracy on a few tasks from CIFAR100 (each task is a superclass). Each cell represents a different experiment, i.e, there is no continual learning being performed here. Cells are colored warm if accuracy is worse than the median accuracy of that row. For instance, multi-task training with 11 tasks is beneficial for "Man-made Outdoor" but accuracy drops drastically upon introducing task #12, it improves upon introducing #14, while task #17 again leads to a drop. One may study the other rows to reach a similar conclusion: there is non-trivial competition between tasks, even in commonly used datasets. As we show, tackling this effectively is the key to obtaining good performance on multi-task learning problems. See Appendix B.1 for a more elaborate version.

Let the set of tasks considered at episode $k$ be denoted by $\bar{P}_k = \{P_{\omega_k^1}, \ldots, P_{\omega_k^b}\}$ where $b \leq k$ is a hyper-parameter and $\omega_k^i \in \{1, \ldots, k\}$. Training on $\bar{P}_k$ will involve, like the example above, training one model with a feature generator $h_k$ and task-specific classifiers $g_{k,\omega_k^i}$ for each task selected in that round. Such models, one trained in each round, together form the "Model Zoo". After $k$ rounds, data from, say, $P_i$ with $i \leq k$ can be predicted using the average of class probabilities output by all models that were fitted on that task, i.e.,

$$p_{k,i}(y \mid x) \propto \sum_{l=1}^{k} \mathbf{1}_{\{P_i \in \bar{P}_l\}} \, g_{l,i} \circ h_l(x). \qquad (7)$$

This expression is also used to predict at test time.

**Selecting tasks to train with for each round using boosting.** In principle, we could use the transfer exponents $\rho_{ij}$ to select synergistic tasks, but computing the transfer exponents is essentially as difficult as training on all tasks, a continual learning does not have access to all tasks *a priori*. We therefore develop an automatic way to select tasks in each round. We draw inspiration from boosting (Schapire

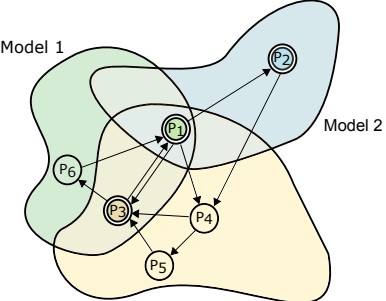

**Figure 3:** Ideally, we want to train synergistic tasks together, e.g., Model 1 for $P_1$ using $P_3, P_6$ and Model 3 for $P_3$ using $P_1, P_4, P_5$. At test time, all models (1, 2, 3) that were trained on a particular task, say $P_1$ would make predictions. Model Zoo is a simple, scalable instantiation of this idea. Discovering noncompeting tasks is difficult, so it selects tasks that have high training loss under the current ensemble.

and Freund, 2013) for this purpose. Recall the AdaBoost algorithm which builds an ensemble of weak learners (they can be any learner in principle Mason et al. (1999)), each of which is fitted upon iteratively re-weighted training data (Breiman, 1998). We think of the models learned at each episode of continual learning in Model Zoo as the "weak learners" and each round of boosting as the equivalent of each episode of continual learning. Let $\bar{w}_k \in \mathbb{R}^n$ be a normalized vector of task-specific weights. After episode $k$

$$\bar{w}_{k,i} \propto \exp\left(-1/m \sum_{(x,y) \in S_i} \log p_{k,i}(y \mid x)\right). \qquad (8)$$

for each task $P_i$ with $i \leq k$; for $i > k$, $\bar{w}_{k,i} = 0$. Tasks for the next round $\bar{P}_{k+1}$ are drawn from a multinomial distribution with weights $\bar{w}_k$. Therefore, tasks with a low empirical risk under the current Model Zoo get a low weight for the next boosting round. Just like AdaBoost drives down the training error on *all* samples to zero exponentially (Schapire and Freund, 2013) by iteratively focusing upon difficult-to-classify samples, Model Zoo achieves a low empirical risk on *all* tasks as more models are added.

**The key feature of Model Zoo** is that it *automatically* splits the capacity across sets of tasks. Even if competing tasks are chosen in one round, which may result in high excess risk on some task, it will be chosen again in future rounds if it has a large error under the ensemble. Colloquially speaking, the ensemble in Model Zoo represents a "brain" that grows its learning capacity continually as more tasks are shown to it.

**Remark 6 (Assumptions in the formulation of Model Zoo).** We assume that, both at training time and test time, the identity of the task is known to the continual learner. Data from past tasks is also

stored with the task identity. This is known as the task-incremental setting in the literature (Van de Ven and Tolias, 2019). Recent work in continual learning also studies settings where such task identity is not known, e.g., (Kaushik et al., 2021), Model Zoo is not designed to handle such settings.

## 4 EMPIRICAL VALIDATION

### 4.1 SETUP

**Datasets.** [*] We evaluate on Rotated-MNIST (Lopez-Paz and Ranzato, 2017), Split-MNIST (Zenke et al., 2017), Permuted-MNIST (Kirkpatrick et al., 2017), Split-CIFAR10 (Zenke et al., 2017), Split-CIFAR100 (Zenke et al., 2017), Coarse-CIFAR100 (Rosenbaum et al., 2017; Yoon et al., 2019; Shanahan et al., 2021) and Split-miniImagenet (Vinyals et al., 2016; Chaudhry et al., 2019b). Split-MNIST, Split-CIFAR10, Split-CIFAR100 and Split-miniImagenet use consecutive groups of labels (2, 2, 5 and 10, respectively) to form tasks. Coarse-CIFAR100 is a variant of CIFAR100 where each super-class is considered a different task (Yoon et al., 2019; 2021; Shanahan et al., 2021). Our study in Fig. 2 has found that Coarse-CIFAR100 is a difficult dataset for continual learning, perhaps because of the semantic differences among the different super-classes.

**Neural architectures and training methodology.** We use a small wide-residual network of Zagoruyko and Komodakis (2016) (WRN-16-4 with 3.6M weights) with task-specific classifiers (one fully-connected layer). We also use an even smaller network (0.12M weights) with 3 convolution layers (kernel size 3 and 80 filters) interleaved with max-pooling, ReLU, batch-norm layers, with task-specific classifier layers. Stochastic gradient descent (SGD) with Nesterov's momentum and cosine-annealed learning rate is used to train all models in mixed precision. Ray Tune (Liaw et al., 2018) was used for hyper-parameter tuning using a multi-task learning model on all tasks from Coarse CIFAR-100. When we do full replay, Model Zoo samples $b = \min(k, 5)$ tasks at the $k^{\text{th}}$ episode; for problems with $n = 5$ tasks, we set $b = 2$; note that $b = 1$ indicates no data replay. **All hyper-parameters are kept fixed for all datasets and all experiments (see §4.2).**

See Appendix A for more details.

### 4.2 EVALUATING CONTINUAL LEARNING METHODS

There is a wide variety of problem formulations in the continual learning literature (Farquhar and Gal, 2019a; Prabhu et al., 2020; Vogelstein et al., 2020; Lopez-Paz and Ranzato, 2017; Van de Ven and Tolias, 2019). Formulations vary with respect to whether they allow replaying data from past tasks, the number of epochs the learner is allowed to train each task for, and the capacity of the model being fitted. We next explain these different formulations, the rationale behind them, and how we execute Model Zoo to conform to each of these settings.

(i) The **strict formulation**, e.g., Kirkpatrick et al. (2017); Kaushik et al. (2021), does not allow any replay of data. For the strict formulation of Model Zoo, we simply set $\bar{w}_{k,i} = 0$ for all $i \neq k$ in (8). At each episode, a single model is trained on the current task and added to the zoo—we call this rather simplistic learner **Isolated**. From a practical standpoint, such a formulation imposes a constraint on the amount of computational resources (compute and/or memory) available during training.

(ii) One can **replay data to various degrees**, e.g., all of it (Nguyen et al., 2017; Guo et al., 2020b), or a subset of it (Chaudhry et al., 2019a). Just like AdaBoost, Model Zoo is fundamentally designed to allow full replay of past tasks. However, we can easily execute it with limited replay by only using a subset of the data to compute gradient updates and also the accuracy on past tasks in the $k^{\text{th}}$ episode. We use the nomenclature **Model Zoo (10% replay)** to indicate that only 10% of the data from past tasks is used; algorithms like A-GEM (Chaudhry et al., 2019a) also use 10% of past data on CIFAR100 datasets. See Appendix A.4 for implementation details. Note that Model Zoo without any data replay is simply Isolated. Let us emphasize that across all these problem settings, Model Zoo remains a legitimate continual learner because it gets access to each task sequentially and has a fixed computational budget ($b$ tasks) at each episode. For a multi-task learner, the computational complexity scales with the number of tasks.

(iii) To impose a strict constraint on the computational complexity of each episode some works, e.g., Chaudhry et al. (2019a), train each task for a single epoch. We therefore show results using both **Model Zoo (single epoch)** (where we replay past data for 1 epoch) and **Isolated (single epoch)** (no replay). Even if the rationale behind using each datum only once is well-taken, one single epoch is quite insufficient to train modern deep networks; if one thinks of biological considerations,

---

[*] Some works (Rebuffi et al., 2017; Lopez-Paz and Ranzato, 2017; Chaudhry et al., 2019a; Mirzadeh et al., 2020b) evaluate on a split of the CIFAR100 dataset where each task is random subset of 5 classes. We do not evaluate on this variant because it is difficult to exactly reproduce the composition of tasks; as Fig. 2 suggests different compositions can have vastly different task accuracy. This is also highlighted by large differences in the accuracy on Split-CIFAR100 and Coarse-CIFAR100 in our work.

| Method | Replay | Single-epoch | Rotated-MNIST | Permuted-MNIST | Split-MNIST | Split-CIFAR10 | Split-CIFAR100 | Coarse-CIFAR100 | Split-miniImagenet |
|---|---|---|---|---|---|---|---|---|---|
| EWC (Kirkpatrick et al., 2017) | ✗ | ✗ | •84 | •96.9 | - | - | •42.40 | - | 46.69 |
| GEM (Lopez-Paz and Ranzato, 2017) | ✓ | ✓ | 86.07 | 82.60 | - | - | *67.8 | - | 51.86 |
| A-GEM (Chaudhry et al., 2019a) | ✓ | ✓ | - | 89.1 | - | - | *62.3 | - | 61.13 |
| Stable-SGD (Mirzadeh et al., 2020b) | ✗ | ✓ | 70.8 | 80.1 | - | - | *59.9 | - | 57.79 |
| ER-Reservoir (Chaudhry et al., 2019b) | ✓ | ✓ | - | 79.8 | - | - | *68.5 | - | 64.03 |
| MEGA-II (Guo et al., 2020a) | ✓ | ✓ | - | 91.20 | - | - | 66.12 | - | - |
| RMN (Kaushik et al., 2021) | ✗ | ✗ | - | 97.73 | 99.5 | - | 80.01 | - | - |
| APD (Yoon et al., 2019) | ✗ | ✗ | - | - | - | - | - | 56.81 | - |
| **Our methods** | | | | | | | | | |
| Isolated-small | ✗ | ✗ | - | - | - | 96.88 | 90.18 | 69.07 | 82.48 |
| Model Zoo-small | ✓ | ✗ | - | - | - | 96.85 | 92.06 | 73.72 | 94.27 |
| Model Zoo-small (10% replay) | ✓ | ✗ | - | - | - | 96.58 | 89.76 | 77.18 | 84.6 |
| Isolated-Resnet18-S | ✗ | ✗ | - | - | - | - | 88.95 | - | - |
| Model Zoo-Resnet18-S | ✓ | ✗ | - | - | - | - | 93.15 | - | - |
| Isolated | ✗ | ✗ | 99.64 | 98.03 | 99.98 | 97.46 | 91.90 | 80.72 | 86.28 |
| Model Zoo | ✓ | ✗ | 99.66 | 97.71 | 99.97 | 98.68 | 94.99 | 84.27 | 96.84 |
| Multi-Head (multi-task) | | | 99.66 | 98.16 | 99.98 | 98.11 | 95.38 | 83.19 | 90.83 |

**Table 1: Average per-task accuracy (%) at the end of all episodes.** MNIST, Permuted-MNIST and Rotated-MNIST are not informative benchmarks for judging forward and backward transfer because even Isolated achieves 99%+ accuracy. Model Zoo outperforms, by significant margins, all existing continual learning methods on all datasets. Accuracy of existing methods is worse than Isolated which suggests little to no forward or backward transfer. Model Zoo-small and Isolated-small have comparable number of weights as that of existing methods, and in some cases, much fewer. Model Zoo-Resnet18-S and Isolated-Resnet18-S, make use of the Resnet18-S architecture from Lopez-Paz and Ranzato (2017). Both Model Zoo/Isolated have similar accuracies on Split-CIFAR100 with 3 different architectures and all of them are better than existing methods. This indicates that the improvement in accuracy is not a result of the specific choice of architecture. For single-epoch numbers refer to Fig. 1 and Table 2. **Note:** * indicates that the evaluation was on Split-CIFAR100 with each task containing randomly sampled labels and is hence it is not directly comparable to other methods. All numbers without a marker are from the paper cited in the first column. • denotes that the accuracy is not from the original paper but from one of (Nguyen et al., 2017; Serra et al., 2018; Chaudhry et al., 2019a). Numbers for other methods on Split-MiniImagenet were computed by us using open-source implementations of the original authors.

local-descent algorithms like stochastic gradient descent (SGD) are quite different from recurrent circuits in the biological brain (Kietzmann et al., 2019). We also run single epoch methods using a very small model (0.12M weights); these are **Model Zoo/Isolated-small (single epoch)**.

(iv) **Multi-Head** trains one single model on all tasks to minimize the average empirical risk with task-specific classifiers; mini-batches contain samples from different tasks. Since Multi-Head is trained on all tasks together, it is not a continual learner, but its accuracy is expected to be an upper bound on the accuracy of continual learning methods.

**Evaluation criteria.** We compare algorithms in terms of the validation accuracy averaged across all tasks at the end of all episodes, average per-task forward transfer (accuracy on a new task when it is first seen, larger this number more the forward transfer), average per-task forgetting (gap in the maximal accuracy of a task during continual learning and its accuracy at the end, larger this number more the forgetting and worse the backward transfer), training and inference time, and memory. Let us note that forward transfer is also sometimes called "learning accuracy" (Riemer et al., 2018), and another measure of backward transfer is the gap between the accuracy at the end of training and the initial accuracy of the task.

## 4.3 Results

Table 1 shows the validation accuracy of different continual learning methods on standard benchmark problems. There are many striking observations here.

(i) **Accuracy of existing methods** compared to in Table 1 (see Table A1 as well) **is poorer than Isolated**. This is surprising because Isolated can be thought of as the simplest possible continual learner—one that unfreezes new capacity at each episode and does not replay data. This indicates that existing methods may be failing to achieve forward or backward transfer compared to simply training the task in isolation; Table 2 investigates this further.

(ii) In comparison, **Model Zoo (all three variants**: small, small with 10% data replay and the standard method) **has better accuracy compared to both existing methods as well as Isolated**. This shows the utility of splitting the capacity of the learner across multiple tasks.

(iii) **Model Zoo matches the accuracy of the multi-task learner** in the last row of Table 1 which has access to all tasks beforehand. Surprisingly, **Model Zoo performs better than Multi-Head in spite of being trained in continual fashion**, especially on harder problems like Coarse-CIFAR100 and Split-miniImagenet. This is a direct demonstration of the effectiveness of Model Zoo in mitigating task competition: the capacity splitting mechanism not only avoids catastrophic forgetting, but it can also leverage data from other tasks even if they are shown sequentially.

Table 2 shows a comparison of the methods developed in this paper with existing methods on Split-CIFAR100 in terms of continual-learning specific metrics. We find:

(i) There are no significant differences in the forward transfer performance in the single epoch setting; larger variants of Isolated and Model Zoo do not work well here because a **single epoch is not sufficient to train modern deep networks**. But **Model Zoo and variants show less forgetting**, it is essentially zero. This indicates that although existing methods are designed to avoid forgetting (the single epoch setting aids this directly), say, A-GEM, or EWC, they do forget. Forgetting can be mitigated by the capacity splitting mechanism in Model Zoo. The per-task accuracy of existing methods is also rather low compared to Model Zoo variants.

(ii) If our methods are implemented in the **multi-epoch setting**, then the **forward transfer** is exceptionally good and **almost as good as the average accuracy** of the task. Surprisingly, this does not come at the cost of **forgetting, which is again essentially zero**.

(iii) Even if Model Zoo and its variants are implemented with **very small models** (0.12M weights/episode, which is 2.42M weights/20 episodes), the **accuracy is better** (Table 1). This suggests that Model Zoo is a performant and viable approach to continual learning. In fact, even the larger model used in Model Zoo is a WRN-16-4 with 3.6M weights and therefore we can train multiple models on the same GPU easily; this is why the training time of Model Zoo is about the same as that of Model Zoo-small.

(iv) The simplicity of Model Zoo and its variants results in much smaller training times and comparable inference times as compared to existing methods.

| Method | Inference time (ms/sample) | Training time (min) | Storage Samples (%) | #Weights (M) | Metrics (Multi Epoch) Accuracy (%) | Forgetting (%) | Forward (%) | Metrics (Single Epoch) Accuracy (%) | Forgetting (%) | Forward (%) |
|---|---|---|---|---|---|---|---|---|---|---|
| EWC | 10.34 | 50 | 0 | 1.6 | - | - | - | 42.4 | 17.52 | 67.76 |
| Prog-NN | - | 82 | 0 | 23.7 | - | - | - | 59.2 | 0.0 | 59.2 |
| GEM | 10.34 | 1048 | 5–10 | 1.6 | - | - | - | 61.2 | 6.0 | 67.61 |
| A-GEM | 10.34 | 88 | 5–10 | 1.6 | - | - | - | 62.3 | 7.0 | 70.13 |
| RMN | 2712.4 | - | 0 | 11.5 | 80.01 | - | - | - | - | - |
| **Our methods** | | | | | | | | | | |
| Isolated-small | 2.34 | 17.09 | 0 | 2.42 | 90.18 | 0.0 | 91.18 | 71.6 | 0.0 | 71.6 |
| Model Zoo-small | 11.70 | 31.71 | 100 | 2.42 | 92.28 | 0.17 | 90.0 | 73.67 | 0.20 | 71.91 |
| Model Zoo-small (10% replay) | 11.70 | 22.41 | 10 | 2.42 | 89.76 | 0.22 | 89.8 | 71.09 | 0.69 | 70.5 |
| Isolated | 2.34 | 20.76 | 0 | 54.8 | 91.9 | 0.0 | 91.0 | 50.43 | 0.0 | 50.43 |
| Model Zoo | 31.84 | 41.86 | 100 | 54.8 | 94.99 | 0.21 | 94.02 | 57.67 | 0.81 | 56.58 |

**Table 2:** A comparison of **continual learning evaluation metrics on Split-CIFAR100** for existing methods and the methods developed in this paper. Our methods demonstrate strong forward and backward transfer, high per-task accuracy, smaller training times and comparable inference times. Training times of other methods are from Chaudhry et al. (2019a) and it is the total training time in minutes for all tasks. The Inference time is the per sample prediction latency averaged over 50 mini-batches of size 16. See Appendix A.5 for more details.

| Replay (%) | Split-CIFAR100 | Split-miniImagenet |
|---|---|---|
| 0 | 71.91 | 65.80 |
| 1 | 70.48 | 67.18 |
| 5 | 71.33 | 70.71 |
| 10 | 71.97 | 74.22 |
| 100 | 73.67 | 81.05 |

| # Tasks ($b$) (100% replay) | Split-CIFAR100 | Split-miniImagenet |
|---|---|---|
| 1 | 71.91 | 65.02 |
| 2 | 72.26 | 67.33 |
| 5 | 73.67 | 81.05 |
| 7 | 73.97 | 88.76 |
| 9 | 74.13 | 84.9 |

| Method | Model Zoo | Ensemble of Isolated (100×) |
|---|---|---|
| Split-CIFAR100 | 73.67 | 71.46 |
| Split-miniImagenet | 81.05 | 67.26 |

**Figure 4: Ablation studies** that show the average per-task accuracy as we vary the size of data replay for Model Zoo (left), the number of past tasks sampled at each episode (middle, $b = 1$ implies no replay), and compare Model Zoo with an ensemble of Isolated models (right). These results are for the single-epoch setting and are therefore directly comparable to those in Table 2 and Table 1 as far as comparison to other methods is concerned. Accuracy is roughly the same on Split-CIFAR100 across varying degrees of replay while it improves significantly on Split-miniImagenet; this suggests that Model Zoo also works with very small amounts of data replay. Accuracy on Split-CIFAR100 is consistent as the number of replay tasks is changed but increases on larger datasets like Split-miniImagenet where there are many more tasks. Finally, the performance of Model Zoo is not merely an artifact of ensembling. Even if Isolated is a strong model, a very large ensemble of Isolated compares poorly to Model Zoo with 100% replay; this indicates that Model Zoo can effectively leverage data from past tasks without forgetting. See the Appendix for more ablation studies.

## 5 RELATED WORK

**Theoretical work on learning from multiple tasks.** Works such as Baxter (2000); Maurer (2006), or recent ones like Du et al. (2020); Tripuraneni et al. (2020) study a shared feature generator with task-specific classifiers, and show that the sample complexity of learning a task improves if true task-specific classifiers are diverse enough. It is also appreciated that such a shared feature generator may not exist for dissimilar tasks. So a different perspective on the problem can found in Crammer et al. (2008); Ben-David et al. (2010); Ben-David and Borbely (2008) who show that learning diverse

tasks requires a larger feature generator and, thereby, more samples; we discuss this in §2.2. We build upon Hanneke and Kpotufe (2019; 2020) to construct the transfer exponent in §2; their work shows that even in very favorable settings, e.g., when all tasks have the same optimal classifier, having access to a large number of tasks may not help. Model Zoo is strongly influenced from these results and we think of it as essentially a way to circumvent them.

There are a number of algorithmic tools to estimate task relatedness, e.g., Evgeniou et al. (2005); Cavallanti et al. (2010); Kumar and Daume III (2012), and although such methods are popular in transfer learning (Pentina and Lampert, 2015; Jaakkola and Haussler, 1999), one cannot apply them in continual learning because we do not know the tasks beforehand. As §2 shows, task relatedness is critical to good learning. So, taking inspiration from AdaBoost (Schapire and Freund, 2013), Model Zoo uses a simple indicator of which past tasks can benefit from future ones, these are the ones with low accuracy under the current ensemble.

**Catastrophic forgetting.** has been the focus of a number of continual learning techniques, e.g., episodic memory-based ones (Lopez-Paz and Ranzato, 2017; Chaudhry et al., 2019a; Farajtabar et al., 2020; Guo et al., 2020a), data replay (Robins, 1995; Shin et al., 2017; Lee et al., 2017), new architectures (Serra et al., 2018), generative replay-based (Mocanu et al., 2016; Shin et al., 2017; Liu et al., 2020; Ven et al., 2020), ensemble-based (Aljundi et al., 2017; Wen et al., 2020) and methods that select locally-redundant directions in the weight space (Kirkpatrick et al., 2017; Aljundi et al., 2018; Mallya et al., 2018; Zenke et al., 2017; Chaudhry et al., 2018). Variational methods, e.g., (Nguyen et al., 2017; Farquhar and Gal, 2019b), sequentially update a posterior over the weights and have an elegant foundation in Bayesian methods but implementing them for large datasets remains a challenge. In spite of intense activity, an effective solution to forgetting remains largely unknown.

Model Zoo embraces the fact that forgetting is a fundamental phenomenon of learning multiple tasks and therefore splitting the capacity may be essential; our results indicate that this approach is effectively at tackling forgetting. This approach also significantly improves other key metrics, e.g., forward-backward transfer and computational complexity of training and inference that have received limited attention (Díaz-Rodríguez et al., 2018). Let us note that Model Zoo is designed for the task-incremental continual learning setting (Van de Ven and Tolias, 2019).

**Parameter sharing/isolation.** A single shared feature generator (i.e., hard parameter sharing) is a popular architecture (Kirkpatrick et al., 2017; Lopez-Paz and Ranzato, 2017; Rebuffi et al., 2017; Nguyen et al., 2017; Mirzadeh et al., 2020b; Chaudhry et al., 2019b). It has been recognized that this is not sufficient; this has given rise to methods for soft-parameter sharing that either design or learn specialized routing architectures (Rosenbaum et al., 2017; Sun et al., 2019; Fernando et al., 2017; Devin et al., 2017; Misra et al., 2016; Vandenhende et al., 2019). Model Zoo is a very simplistic instantiation of parameter isolation, or growing (Rusu et al., 2016; Mallya and Lazebnik, 2018; Xu and Zhu, 2018). Model Zoo trains on one episode and never updates the model again but its accuracy does play a role in determining whether a *new* model should be used for that past task, or not. To extend the analogy, just like soft-parameter sharing architectures use, say gradient conflict (Aljundi et al., 2018) or attention (Serra et al., 2018), to determine which synapses to share, Model Zoo uses the training loss of the ensemble to decide what task the new model should be trained upon. It is remarkable that such a simple instantiation performs so well; this has not been noticed before.

## 6 DISCUSSION

Continual learning is an important problem as deep learning systems transition from the traditional paradigm of having a fixed model that makes inferences on user queries to settings where we would like to update the model to handle new types of queries. The key desiderata of such a system are clear: it must display high per-task accuracy and strong forward-backward transfer. This paper seeks to develop such a continual learner and investigates the problem using the lens of task relatedness. It argues that the learner must split its capacity across sets of tasks to mitigate competition between tasks and benefit from synergies among them. We develop Model Zoo, which is a continual learning algorithm inspired by AdaBoost, that grows an ensemble of models, each of which is trained on data from the current episode along with a subset of past tasks. **We show that across a wide variety of datasets, problem formulations, and evaluation criteria, Model Zoo and its variants outperform existing continual learning methods.** We also show that a simple baseline method, where a separate, small model is trained independently in each episode, outperforms a number of existing continual methods. Appendix D discusses these results further.

## 7 Ethics Statement

This work does not raise any ethical concerns.

## 8 Reproducibility Statement

To ensure the reproducability of our work, the full source code is available at `https://github.com/rahul13ramesh/modelzoo_continual`. The code contains scripts to download all datasets, implementations of different variants of our method discussed in the paper, scripts for hyper-parameter tuning, and data analysis. We note here that we use the same set of hyper-parameters for all our experiments; this will help other researchers reproduce our results.

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

## A  DETAILS OF THE EXPERIMENTAL SETUP

### A.1  DATASETS

We performed experiments using the following datasets.

1. Rotated-MNIST (Lopez-Paz and Ranzato, 2017) uses the MNIST dataset to generate 5 different 10-way classification tasks. Each task involves using the entire MNIST dataset rotated by 0, 10, 20, 30, and 40 degrees, respectively.

2. Permuted-MNIST (Kirkpatrick et al., 2017) involves 5 different 10-way classification tasks with each task being a different permutation of the input pixels. The first task is the original MNIST task as is convention. All other tasks are distinct random permutations of MNIST images.

3. Split-MNIST (Zenke et al., 2017) has 5 tasks with each task consisting of 2 consecutive labels (0-1, 2-3, 4-5, 6-7, 8-9) of MNIST.

4. Split-CIFAR10 (Zenke et al., 2017) has 5 tasks with each task consisting of 2 consecutive labels (airplane-automobile, bird-cat, deer-dog, frog-horse, ship-truck) of CIFAR10.

5. Split-CIFAR100 (Zenke et al., 2017) has 20 tasks with each task consisting of 5 consecutive labels of CIFAR100. See the original paper for the exact constitution of each task.

6. Coarse-CIFAR100 (Rosenbaum et al., 2017) has 20 tasks with each task consisting of 5 labels. The tasks are based on an existing categorization of classes into super-classes (https://www.cs.toronto.edu/ kriz/cifar.html).

7. Split-miniImagenet (Vinyals et al., 2016) is a ariant introduced in Chaudhry et al. (2019b), consisting of 20 tasks, with each task consisting of 10 consecutive labels. We merge the meta-train and meta-test categories to obtain a continual learning problem with 20 tasks. Each task containing 10 consecutive labels and 20% of the samples are used as the validation set.

The CIFAR10 and CIFAR100-based datasets consist of RGB images of size $32 \times 32$ while MNIST-based datasets consist of images of size $28 \times 28$. The Mini-imagenet dataset consists of RGB images of size $84 \times 84$.

### A.2  ARCHITECTURE

We use the Wide-Resnet (Zagoruyko and Komodakis, 2016) architecture for some of our experiments. The final pooling layer is replaced with an adaptive pooling layer in order to handle input images of different sizes. Convolutional layers are initialized using the Kaiming-Normal initialization. The bias parameter in batch normalization is set to zero with the affine scaling term set to one. The bias of the final classification layer is also set to zero; this helps keep the logits of the different tasks on a similar scale.

To ensure that the number of weights is similar to those in other methods, we also consider a smaller convolution neural network consisting of 3 convolution layers, with batch-normalization, ReLU and max-pooling present between each layer.

### A.3  TRAINING SETUP

**Optimization.** All models are trained in mixed-precision (32-bit weights, 16-bit gradients) using Stochastic Gradient Descent (SGD) with Nesterov's acceleration with momentum coefficient set to 0.9 and cosine annealing of the learning rate schedule for 200 epochs. Training of any model with multiple tasks involves mini-batches that contain samples from all tasks.

**Hyper-parameter optimization.** We used Ray Tune (Liaw et al., 2018) for hyper-parameter optimization. The Async Successive Halving Algorithm (ASHA) scheduler (Li et al., 2018) was used to prune hyper-parameter choices with the search space determined by Nevergrad (Rapin and Teytaud, 2018). The mini-batch size was varied over [8, 16, 32, 64]; the logarithm (base 10) of the learning rate was sampled from a uniform distribution on $[-4, -2]$; dropout probability was sampled from a uniform distribution on $[0.1, 0.5]$; logarithm of the weight decay coefficient was sampled from $[-6, -2]$. We used a set of experiments for continual learning on the Coarse-CIFAR100 dataset with different samples/class (100 and 500) to perform hyper-parameter tuning.

**The final values of training hyper-parameters** that were chosen are, learning-rate of 0.01, mini-batch size of 16, dropout probability of 0.2 and weight-decay of $10^{-5}$.

Model Zoo uses $\ell = \min(k, 5)$ at each round of continual learning where $n$ is the number of tasks; for tasks with only $5$ tasks (MNIST-variants) we use $\ell = 2$. We did not tune these two hyper-parameters using Ray because it is quite cumbersome to do so. We selected these values manually across a few experiments; changing them may result in improved accuracy for Model Zoo.

**All hyper-parameters are kept fixed for all datasets, architectures, and experimental settings.** . We are interested in characterizing the performance of Model Zoo and its variants across a broad spectrum of problems and datasets. While we believe we can get even better numerical accuracy, by tuning hyper-parameters specially for each problem, we do not so for the sake of simplicity. As the main paper discusses, we outperform existing methods quite convincingly across the board in both multi-task and continual learning.

**Data augmentation.** MNIST and CIFAR10/100 datasets use padding (4 pixels) with random cropping to an image of size 28×28 or 32×32 respectively for data augmentation. CIFAR10/100 images additionally have random left/right flips for data augmentation. Images are finally normalized to have mean 0.5 and standard deviation 0.25. Split-miniImagenet uses the same augmentation as CIFAR-10 and CIFAR-100. We use augmentation even in the single epoch setting.

### A.4 MODEL ZOO WITH LIMITED REPLAY

As discussed in §4.2, this work considers Model Zoo (10%) which stores only 10% of the data from the past tasks, in order to compare to other methods that make use of limited replay. When the task (say task A) is first seen, Model Zoo is allowed to use all available data. For all future episodes, if Model Zoo picks a past task to retrain with, such a retraining uses only a fixed subset of the tasks' data (10% of the samples are selected at random for this purpose). We sample each mini-batch to contain an equal number of samples from all past and current task. At inference time, the member of Model Zoo that is trained on all data of task A (this is the model that was fitted when task A was first shown to the continual learner) is assigned a proportionately larger weight in Eq. (7). For 10% replay, this will amount to 10 × larger weight than other models which used 10% data from task A. Mathematically, both of these training and inference modifications are equivalent to using coefficients that scale up the loss of the past task depending upon the number of samples that it has.

### A.5 EVALUATING TRAINING AND INFERENCE TIMES

In this section, we describe the methodology used to estimate training and inference times reported in Table 2.

**Inference time.** The column titled inference time corresponds to per-sample prediction latency in milli-seconds. All entries for inference time in Table 2 were computed by us on an Nvidia V100 GPU and therefore they can be compared directly with each other. Note that inference times can be computed using only the architecture built by each method at the end of all continual learning episodes. We obtained the architectures used in each method from open-source implementations of the original authors (https://github.com/facebookresearch/agem and https://github.com/imirzadeh/stable-continual-learning). Inference time is computed by processing 50 mini-batches from CIFAR-100, each of batch-size 16. The inference time is computed by normalizing the total computation time by (size of mini-batch × number of mini-batches), which gives the average inference-time per sample. For Model-Zoo, we assume that inference time is approximately $b = 5$ times the inference time of Isolated, where $b$ is number of tasks sampled in every round).

**Training time.** corresponds to the time (in minutes) required to train all episodes of the Split-CIFAR100 dataset. Establishing an accurate comparison is difficult because different papers used different hardware but we have strived to be fair. The training time for EWC, Prog-NN, GEM and A-GEM are obtained from Chaudhry et al. (2019a) (we divide the numbers by 5 since this paper reports the sum of training times of 5 different runs). Chaudhry et al. (2019a) also report the training time for naive fine-tuning (21 mins) which in theory, should be very similar to the training time of our Isolated learner (the training time for us is 20.76 mins on one V100 GPU). Since the two numbers are quite similar, we can estimate training time of the other continual learning methods using their computational cost relative to naive fine-tuning. Therefore, the estimate of the training times that we have reported in Table 2 can be compared to each other.

## B ADDITIONAL EXPERIMENTS

### B.1 UNDERSTANDING TASK COMPETITION

To understand which tasks aid each other's learning and which compete for capacity and may thereby deteriorate performance, we investigated the Coarse-CIFAR100 dataset extensively. We first computed the pairwise task competition by comparing the relative gain/drop in classification accuracy of each pair of tasks when the row task is trained in isolated versus training the row and column tasks together using a simple multi-task learner (Multi-Head). Fig. A1 discusses the results.

Fig. A2, is the extended version of Fig. 2. It shows the validation accuracy of each task (along a single row) as more tasks are added to Multi-Head. Each column is a single Multi-Head model trained on a subset of tasks from scratch. As more tasks are added, the accuracy of most tasks increases.

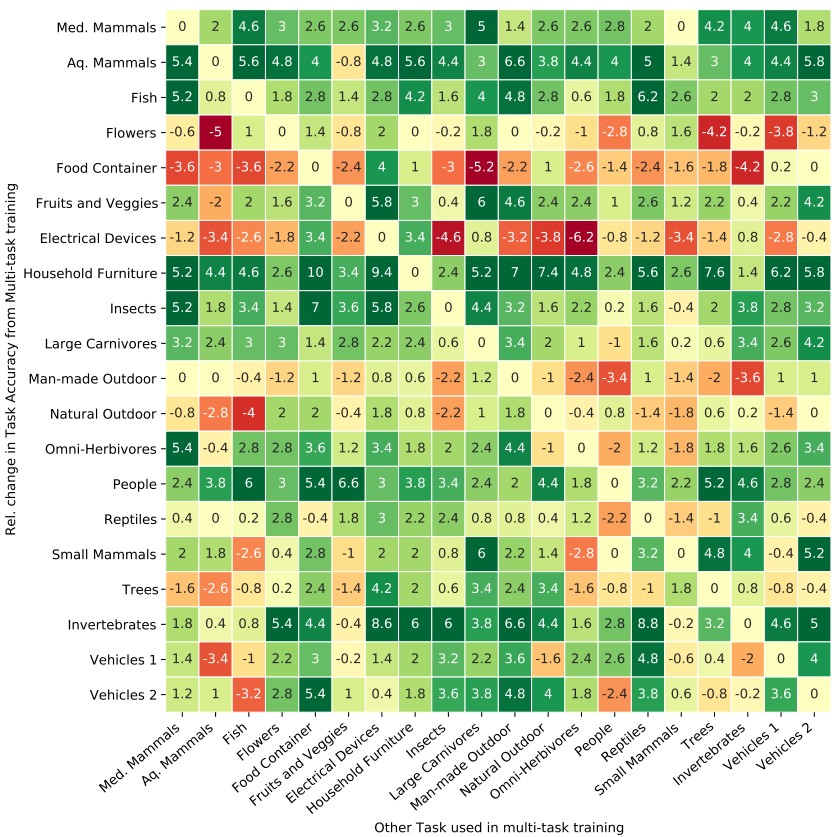

**Figure A1: Pairwise task competition matrix**. Cells are colored by the gain(green)/loss(warm) of accuracy of pairwise Multi-Head training as compared to training the row-task in isolation; this is a good proxy for the transfer coefficient $\rho_{ij}$ in (5). Although most pairs benefit each other (green), certain tasks, e.g., "Food Container" are best trained in isolation while others such as "Aquatic Mammals" are typically detrimental to most other tasks. One can study this matrix and identify many more such properties. In summary, whether tasks aid or hurt each other is quite nuanced even for CIFAR100.

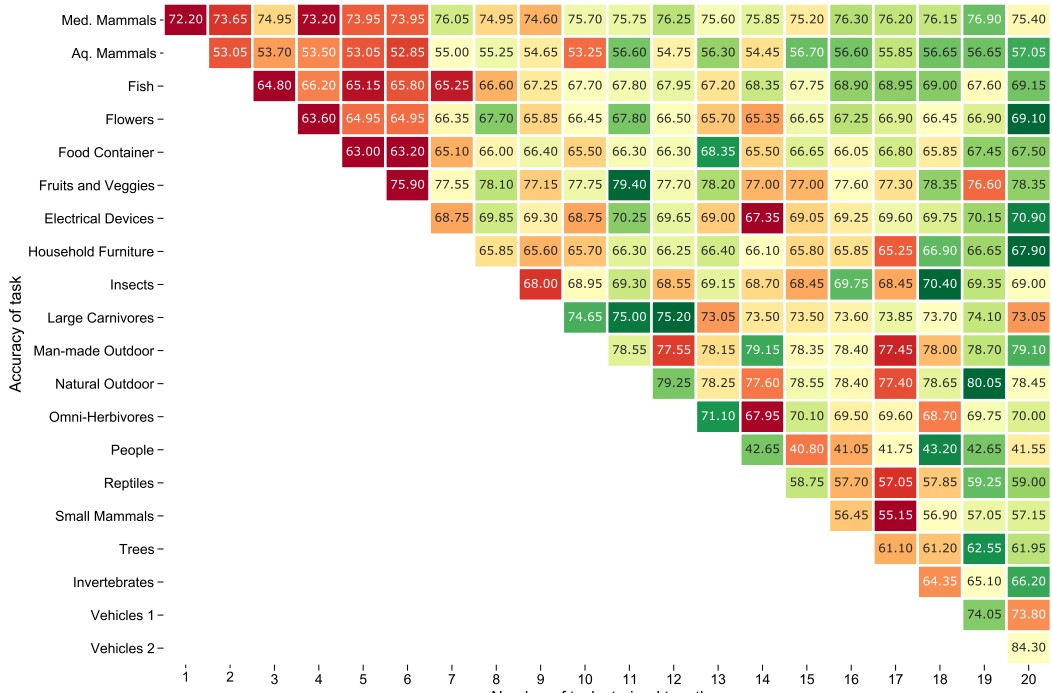

**Figure A2:** In order to demonstrate how some tasks help and some tasks hurt each other, we run Multi-Head for a varying number of tasks (X-axis) and track the accuracy on a few tasks from Coarse-CIFAR100. The order of tasks is the same for rows (top to bottom) and the columns (left to right). In other words, the first cell (the diagonal) indicates the accuracy of the task trained by itself in isolation (Isolated). Cells are colored warm if accuracy is worse than the median accuracy of that row. For instance, multi-task training with 11 tasks is beneficial for "Man-made Outdoor" but accuracy drops drastically upon introducing task #12, it improves upon introducing #14, while task #17 again leads to a drop. One may study the other rows to reach a similar conclusion: there is non-trivial competition between tasks, even in commonly used datasets. Tackling this issue effectively is the key to obtaining good performance on multi-task learning problems

However, the increase is not monotonic with each added task, and if one follows a particular row, there are non-trivial patterns wherein adding a particular task may deteriorate the performance on the row task and adding some other task later may recover the lost accuracy. This is a direct demonstration of the tussle between the task competition term (first) and the concentration term (third) in Theorem 2. This indicates that training on the appropriate set of tasks is crucial to learn from multiple tasks.

## B.2 Competition between tasks of typical benchmark datasets

Next, we investigated such task competition on other continual learning datasets, namely, Permuted-MNIST, Rot-MNIST, Split-CIFAR10, and Split-MNIST. It is clear from Fig. A3 that there is very little competition in this case. Either the tasks are quite different from each other (like the case of Permuted-MNIST), or they are synergistic (most cells are green), or they do not hurt each other's performance, i.e., they may correspond to the model in §2.2. Note that Rotated-MNIST exactly corresponds to the multi-view setting discussed in §2.2 were different input images are simple transformations of each other.

## B.3 Visualizing successive iterations of Model Zoo

In order to understand how the accuracy of Model Zoo evolves on all tasks as a function of the episodes, we created Fig. A4. This is a very insightful picture and we can draw the following conclusions from it.

  (i) The accuracy along the diagonal of most tasks increases along the row, i.e., across episodes. Only for a few tasks like Food Container the accuracy drops in later episodes. Note that we also see from Fig. A1 that Food Container is a task that is best trained in isolation because it leads to deterioration of accuracy when trained with essentially any other task.

  (ii) The is strong backward transfer throughout the dataset, i.e., the accuracy of a task shown in earlier rounds increases, sometimes a large number, as later synergistic tasks are shown to the learner.

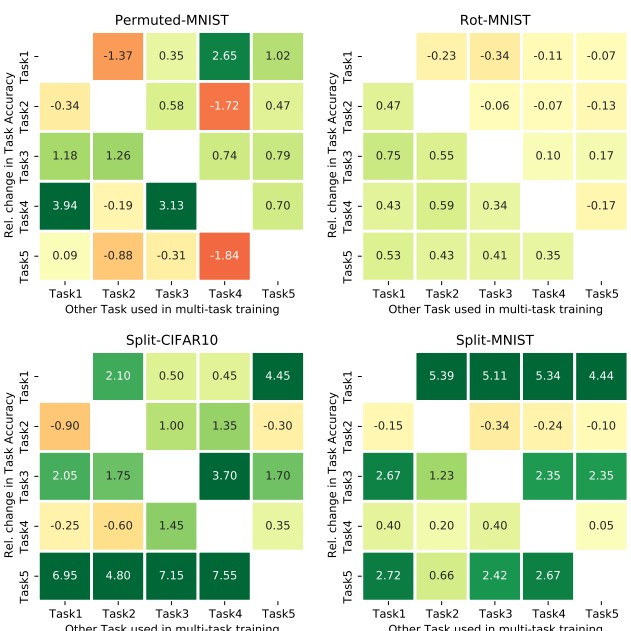

**Figure A3:** Each row is the relative increase/decrease (green/red) in accuracy of a two task Multi-Head learner compared to Isolated trained on the task corresponding to the particular row; all entries are computed using 100 samples/class. Cells are colored green for accuracy gained, and warm for accuracy dropped; the entries in this matrix are a good proxy for the transfer coefficient $\rho_{ij}$ in (5). A similar plot for Coarse-CIFAR100 tasks is shown in the right panel of Fig. 2. Split-CIFAR10 and Split-MNIST indicate that most tasks mutually benefit each other. This is also true, but to a lesser extent, for Rotated-MNIST. Permuted-MNIST is a qualitatively different problem than these, perhaps because there is no obvious relationship between the tasks and there exist some tasks that lead to a large deterioration of accuracy.

(iii) We also see strong forward transfer. Roughly speaking, in the second half of the rows, the initial accuracy of most tasks does not improve much with successive episodes. This suggests that these tasks already have a good initial accuracy, i.e., there is good forward transfer in the learner.

We advocate that such plots should be made for different continual learning algorithms to obtain a precise picture of the amount of forward and backward transfer.

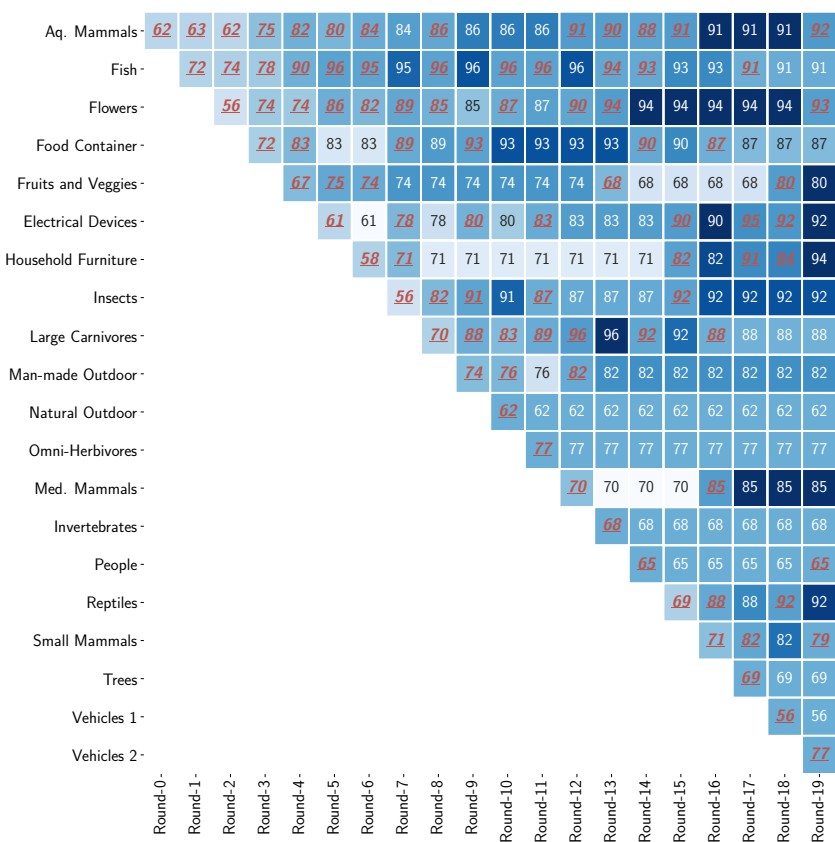

**Figure A4:** The iterations of Model Zoo are visualized for the Split-miniImagenet dataset for 20 rounds, with 5 tasks selected in every iteration of Model Zoo. Red elements are tasks that were selected by boosting in that particular round. We observe that the accuracy of most tasks improves over the rounds, which indicates the utility of Model Zoo-like training scheme This plot also indicates that Model Zoo can improve the per-task accuracy on nearly all tasks. The model is trained for only a single-epch per boosting round.

## B.4 BASELINE PERFORMANCE OF ISOLATED TRAINING ON COARSE-CIFAR100

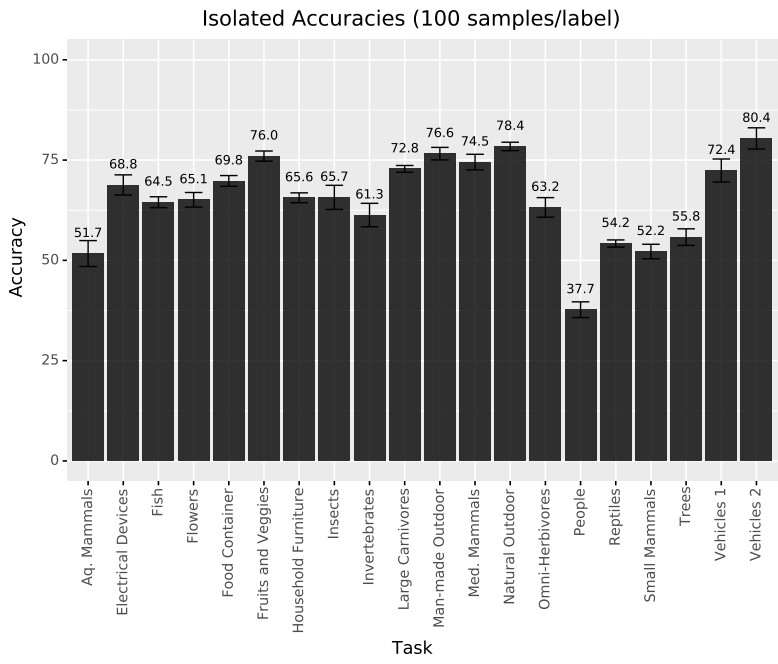

**Figure A5:** Per-task accuracies of Isolated on the Coarse-CIFAR100 dataset for two cases, one with 100 samples/class (top) and another with all 500 samples/class (bottom). Two points are very important to note here. First, there is a large improvement in the two accuracies for all tasks when the learner has access to more samples. Second, different tasks have very different accuracies when trained in isolation (using the same WRN-16-4 model). This indicates that different tasks are very different in terms how hard they are, for some tasks such as People, the base accuracy of the model is quite low and one must have lots of samples in order to perform well. A lot of other multi-task learning datasets, e.g., derivatives of MNIST (or even CIFAR10 to an extent) are unlike CIFAR100 in this respect.

## B.5 ADDITIONAL EXPERIMENTS

Table A1 is a more detailed version of Table 1 in the main paper.

| Method | Replay | Single Epoch | Rotated-MNIST | Permuted-MNIST | Split-MNIST | Split-CIFAR10 | Split-CIFAR100 | Coarse-CIFAR100 | Split-MiniImagenet |
|---|---|---|---|---|---|---|---|---|---|
| GEM (Lopez-Paz and Ranzato, 2017) | ✓ | ✓ | 86.07 | 82.60 | - | - | 67.8* | - | 51.86 |
| A-GEM (Chaudhry et al., 2019a) | ✓ | ✓ | - | 89.1 | - | - | 62.3* | - | 61.13 |
| ER-Reservoir (Chaudhry et al., 2019b) | ✓ | ✓ | - | 79.8 | - | - | 68.5* | - | 64.03 |
| MC-SGD (Mirzadeh et al., 2020a) | ✓ | ✓ | 82.63 | 85.3 | - | - | 63.30 | - | - |
| MEGA-II (Guo et al., 2020a) | ✓ | ✓ | - | 91.20 | - | - | 66.12 | - | - |
| OGD (Farajtabar et al., 2020) | ✗ | ✓ | 88.32 | 86.44 | 98.84 | - | - | - | - |
| Stable-SGD (Mirzadeh et al., 2020b) | ✗ | ✓ | 70.8 | 80.1 | - | - | 59.9* | - | 57.79 |
| TAG (Malviya et al., 2021) | ✗ | ✓ | - | - | - | - | 62.79 | - | 57.2 |
| VCL (Nguyen et al., 2017) | ✓ | ✗ | - | 95.5 | 98.4 | - | - | - | - |
| FRCL (Titsias et al., 2020) | ✓ | ✗ | - | 94.3 | 97.8 | - | - | - | - |
| FROMP (Pan et al., 2020) | ✓ | ✗ | - | 94.9 | 99.0 | - | - | - | - |
| EWC (Kirkpatrick et al., 2017) | ✗ | ✗ | •84 | •96.9 | - | - | •42.40 | - | - |
| Prog-Nets (Rusu et al., 2016) | ✗ | ✗ | - | •93.5 | - | - | •59.2 | - | - |
| SI (Zenke et al., 2017) | ✗ | ✗ | - | •97.1 | •98.9 | - | - | - | - |
| HAT(Serra et al., 2018) | ✗ | ✗ | - | 98.6 | 99.0 | - | - | - | - |
| APD (Yoon et al., 2019) | ✗ | ✗ | - | - | - | - | - | 56.81 | - |
| FedWeIT (Yoon et al., 2021) | ✗ | ✗ | - | - | - | - | - | 55.16 | - |
| RMN (Kaushik et al., 2021) | ✗ | ✗ | - | 97.73 | 99.5 | - | 80.01 | - | - |
| **Our methods** | | | | | | | | | |
| Isolated-small | ✗ | ✗ | - | - | - | 96.88 | 90.18 | 69.07 | 82.48 |
| Model Zoo-small | ✓ | ✗ | - | - | - | 96.85 | 92.06 | 73.72 | 94.27 |
| Model Zoo-small (10% replay) | ✓ | ✗ | - | - | - | 96.58 | 89.76 | 77.18 | 84.6 |
| Isolated-Resnet | ✗ | ✗ | - | - | - | - | 88.95 | - | - |
| Model Zoo-Resnet | ✓ | ✗ | - | - | - | - | 93.15 | - | - |
| Isolated | ✗ | ✗ | 99.64 | 98.03 | 99.98 | 97.46 | 91.90 | 80.72 | 86.28 |
| Model Zoo | ✓ | ✗ | 99.66 | 97.71 | 99.97 | 98.68 | 94.99 | 84.27 | 96.84 |
| Multi-Head (multi-task) | | | 99.66 | 98.16 | 99.98 | 98.11 | 95.38 | 83.19 | 90.83 |

**Table A1:** Average per-task accuracy (%) for continual learning at the end of all episodes. MNIST, Permuted-MNIST and Rotated-MNIST are not informative benchmarks for judging forward and backward transfer because even Isolated achieves 99%+ accuracies. Model Zoo outperforms, by significant margins, all existing continual learning methods; in fact their accuracy is worse than Isolated which suggests little to no forward or backward transfer. **Note:** * indicates that the evaluation was on Split-CIFAR100 with each task containing randomly sampled labels and is hence it is not directly comparable to other methods. All numbers without a marker are from the paper cited in the first column. • denotes that the accuracy is not from the original paper but from one of (Nguyen et al., 2017; Serra et al., 2018; Chaudhry et al., 2019a). Numbers for other methods on Split-MiniImagenet were computed by us using open-source implementations of the original authors.

## B.6    SINGLE EPOCH METRICS

We obtain metrics from publicly available implementations of a few different continual learning algorithms, which are shown in Tables A2 and A3. We see that Model Zoo and its variants uniformly have essentially no forgetting and good forward transfer. The average per-task accuracy is also higher than existing methods on these datasets. These tables show results for single-epoch training (to be consistent with the implementation of these existing methods).

| Method | Avg. Accuracy | Forgetting | Forward |
|---|---|---|---|
| SGD | 34.52 | 19.88 | 53.30 |
| EWC | 34.71 | 18.60 | 52.19 |
| AGEM | 37.23 | 16.96 | 52.72 |
| ER | 41.36 | 14.29 | 54.87 |
| Stable-SGD | 37.27 | 12.07 | 48.43 |
| TAG | 43.33 | 12.39 | 55.1 |
| Isolated-small | 58.719 | 0.0 | 58.71 |
| Model Zoo-small | 60.3 | 0.370 | 59.13 |
| Isolated-large | 41.28 | 0.0 | 41.28 |
| Model Zoo-large | 46.98 | 0.38 | 44.43 |

**Table A2:** Single Epoch continual learning metrics on Coarse-CIFAR100

| Method | Avg. Accuracy | Forgetting | Forward |
|---|---|---|---|
| SGD | 46.69 | 16.653 | 62.35 |
| EWC | 47.93 | 14.26 | 61.34 |
| AGEM | 51.86 | 10.102 | 61.13 |
| ER | 55.41 | 9.52 | 64.03 |
| Stable-SGD | 49.28 | 9.76 | 57.79 |
| TAG | 58.38 | 5.15 | 63.00 |
| Isolated-small | 65.8 | 0.0 | 65.8 |
| Model Zoo-small | 81.049 | 1.278 | 66.57 |
| Isolated-large | 40.2 | 0.0 | 40.25 |
| Model Zoo-large | 64.12 | 0.27 | 48.34 |

**Table A3:** Single Epoch continual learning metrics on Split-MinImagenet

## B.7 TRACKING INDIVIDUAL TASK ACCURACIES

We next study how the individual per-task accuracy evolves on different datasets. The following figures are extended versions of the right panel of Fig. 1. We see that the accuracy of all tasks increases with successive episodes. This is quite uncommon for continual learning methods and indicates that Model Zoo essentially does not suffer from catastrophic forgetting. We have also juxtaposed the corresponding curves of the single-epoch setting with the multi-epoch training in Model Zoo; we would like to demonstrate the dramatic gap in the accuracy of these problem settings. Even if single-epoch variant of Model Zoo also does not forget (its accuracy is much better than existing continual learning methods), the multi-epoch variant has much higher accuracy for every task. This indicates that continual learning algorithms should also focus on per-task accuracy in addition to mitigating forgetting, if they are to be performant. The performance of Model Zoo is evidence that we can build effective continual learning methods that do not forget.

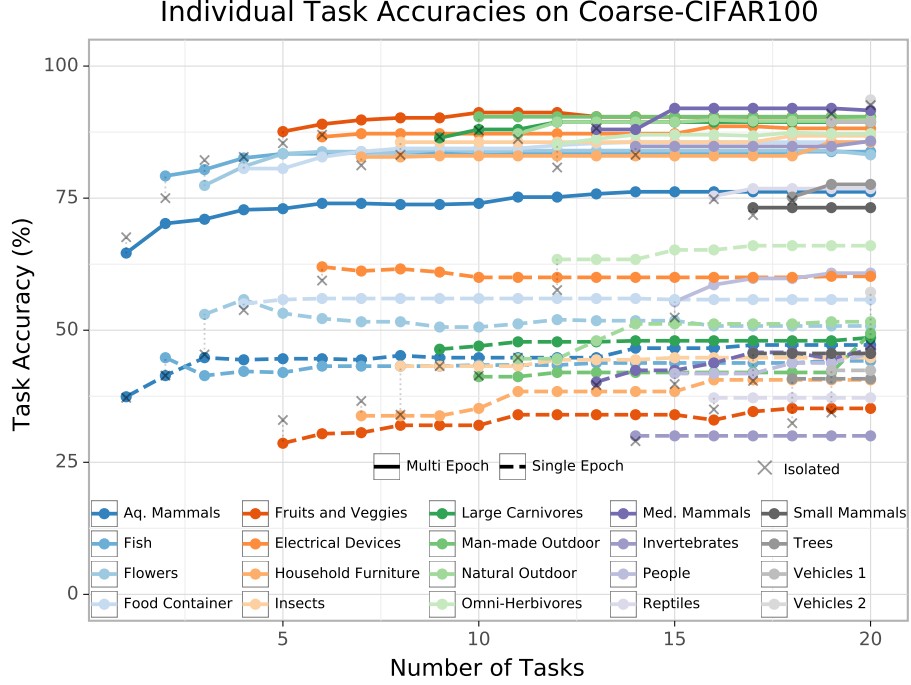

**Figure A6:** Evolution of task accuracy on Coarse-CIFAR100

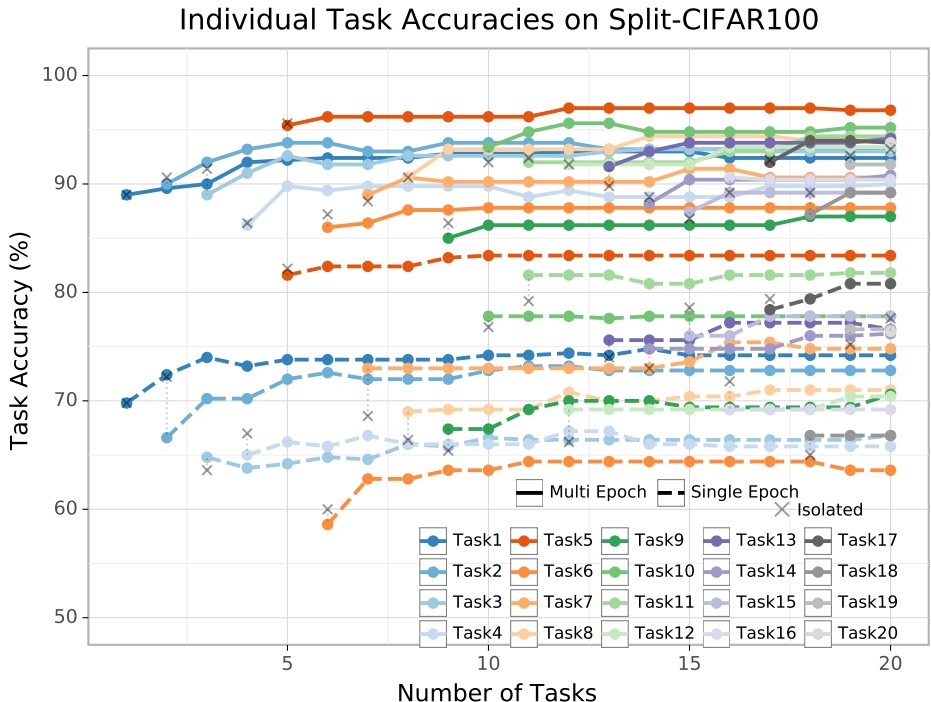

**Figure A7:** Evolution of task accuracy on Split-CIFAR100

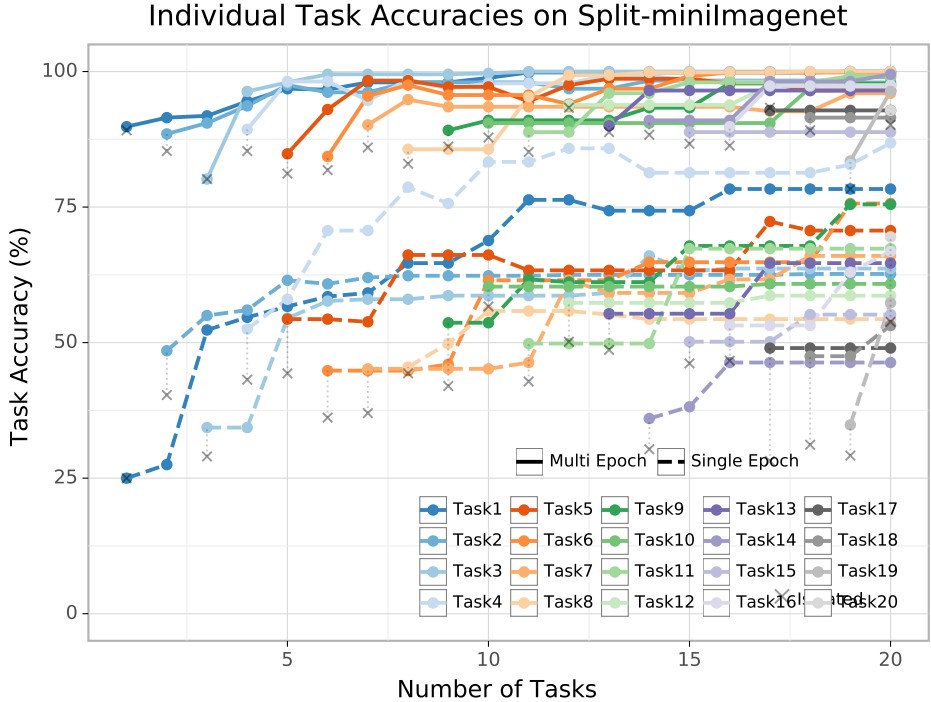

**Figure A8:** Evolution of task accuracy on Split-miniImagenet

## B.8  Comparison To Existing Methods

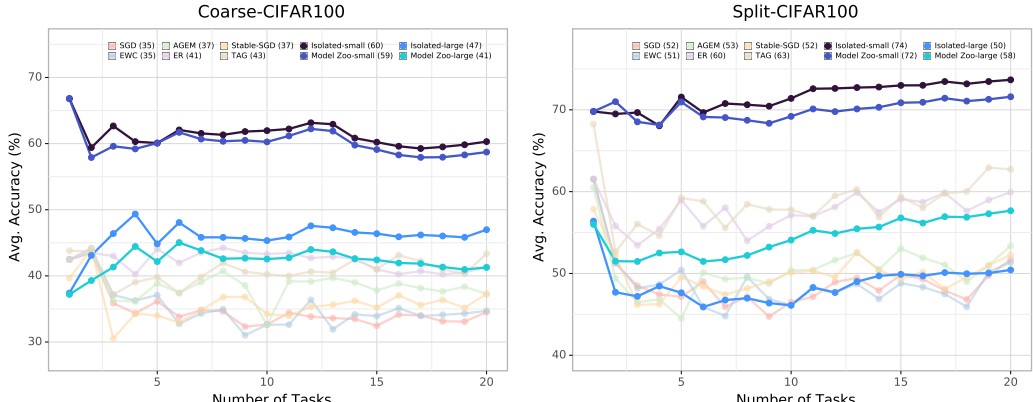

**Figure A9:** This figure compares Model Zoo to existing continual learning methods on the Coarse-CIFAR100 and Split-CIFAR100 datasets with respect to average task accuracy. Model Zoo and its variants are in bold, similar to the left panel of Fig. 1 (which is for Split-miniImagenet). Isolated-small and Model Zoo-small significantly outperform existing methods. All methods in the figure are run in the single-epoch setting.

## B.9  Additional Continual Learning Experiments on 100 samples/label

We also performed continual learning experiments with 100 samples/class in Table A4. We find that Model Zoo-continual obtains an accuracy that lies in between those of Isolated and the approximate upper bound given by Multi-Head (multi-task learning). Note that we have shown that matching or improving upon the performance of Isolated (which trains a model independently for each task) for continual learning is quite difficult because it necessitates effective forward-backward transfer. Doing so indicates strong ability of the learner for *both* forward and backward transfer. In some cases, the continual learner even outperforms Multi-Head trained on all tasks together. This table indicates that Model Zoo can be used as a continual learning and demonstrate nontrivial forward and backward transfer even with few samples from each class.

| Dataset | Isolated | Multi-Head (multi-task) | Model Zoo-Continual |
|---|---|---|---|
| Rotated-MNIST | $98.17 \pm 0.24$ | $98.47 \pm 0.18$ | $98.44 \pm 0.17$ |
| Split-MNIST | $97.11 \pm 1.21$ | $99.47 \pm 0.08$ | $98.98 \pm 0.51$ |
| Permuted-MNIST | $84.59 \pm 1.65$ | $86.36 \pm 1.15$ | $86.04 \pm 1.68$ |
| Split-CIFAR10 | $82.09 \pm 0.76$ | $85.73 \pm 0.60$ | $84.17 \pm 0.60$ |
| Split-CIFAR100 | $80.04 \pm 0.44$ | $87.93 \pm 0.50$ | $86.27 \pm 0.19$ |
| Coarse-CIFAR100 | $65.34 \pm 0.41$ | $69.05 \pm 0.38$ | $66.80 \pm 6.27$ |

**Table A4:** Average per-task accuracy (%) for continual learning at the end of all episodes using 100 samples/class, bootstrapped across 5 datasets (mean $\pm$ std. dev.). Model Zoo-continual performs better than Isolated on all problems even if tasks are shown sequentially.

We next visualize the evolution of the per-task test accuracy for various datasets. This is a qualitative way to investigate forward and backward transfer in the learner. Forward transfer is positive if the accuracy of a newly introduced task in a particular episode is higher than what it would be if the task were trained in isolation. Backward transfer is positive if successive episodes and tasks result in an increase in the accuracy of tasks that were introduced earlier in continual learning. Both Appendix B.7 and Fig. A10 consistently show non-trivial forward and backward transfer.

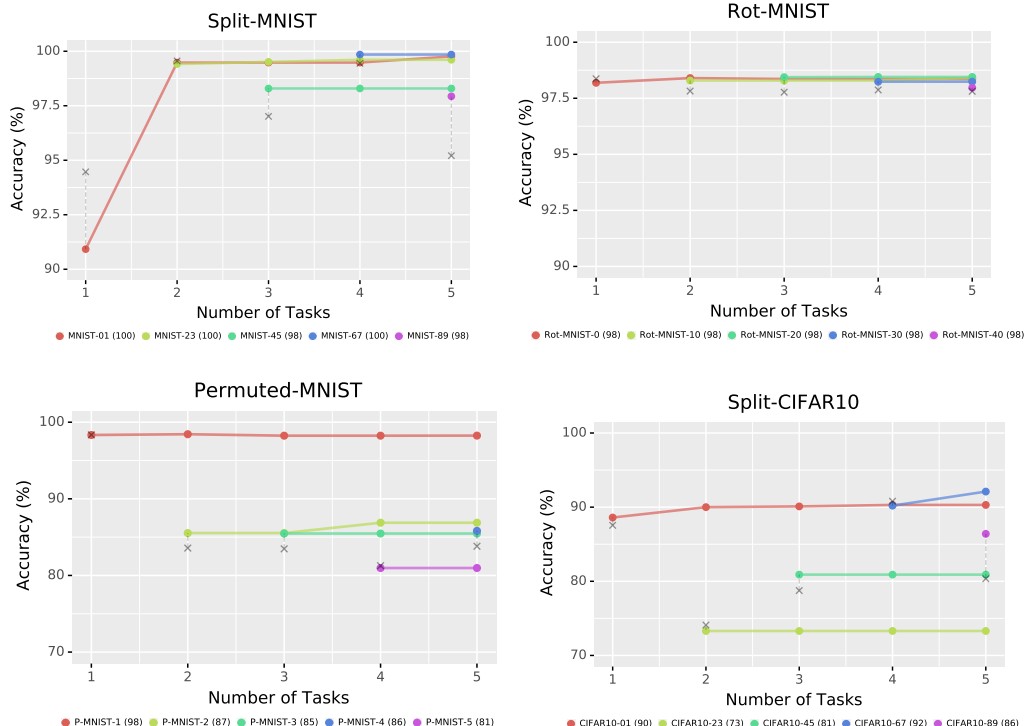

**Figure A10:** Per-task validation accuracy as a function of the number of episodes of continual learning for problems using variants of CIFAR10 and MNIST datasets using Model Zoo-continual. Each task has 100 samples/class. X-markers denote accuracy of Isolated on the new task. We see both forward transfer (Model Zoo often starts with a higher accuracy than Isolated) and backward transfer (accuracy of some past tasks improves in later episodes). For problems like Permuted-MNIST and Rotated-MNIST, there is little forward or backward transfer.

### B.10 MODEL ZOO WITH UNIFORM SAMPLING

At each round of boosting, Model Zoo samples tasks according to equation (8) i.e., tasks with high loss under the current ensemble have a higher probability of being selected in the next round. To study the importance of this heuristic, we compare Model Zoo to a variant called Model Zoo (uniform). Model Zoo (uniform) samples uniformly over all seen tasks for each round, as opposed to using equation (8).

Table A5 compares the accuracy of Model Zoo and Model Zoo (uniform) on the Coarse-CIFAR100 dataset. Model Zoo is marginally better than Model Zoo (uniform) indicating that using the training loss is a cheap proxy for splitting the capacity amongst related tasks. At the same time, this also indicates that a better measure of task-distances can further improve performance.

| Method | Avg. Accuracy |
|---|---|
| Model Zoo | 84.27 |
| Model Zoo (uniform) | 83.60 |

**Table A5:** Comparison of accuracies on the Coarse-CIFAR100 dataset

## C PROOFS

**Proof of Theorem 2**. From the definition of $\rho_{ij}$ relatedness for tasks, we have

$$c\,\mathcal{E}_{P_i}^{1/\rho_{i1}}(h) \geq \mathcal{E}_{P_1}(h, h_i^*)$$
$$= \mathcal{E}_{P_1}(h) - \mathcal{E}_{P_1}(h_i^*, h_1^*).$$

for any $i, j \leq n$ and $h \in H$. Let us denote $\rho_{(i)} = \rho_{i1}$. We can sum over $i \in \{1, \ldots, k\}$ and divide by $k$ to get

$$\mathcal{E}_{P_1}(h) \leq \frac{1}{k}\sum_{i=1}^{k}\mathcal{E}_{P_1}(h_{(i)}^*) + \frac{c}{k}\sum_{i=1}^{k}\mathcal{E}_{P_{(i)}}^{1/\rho_{(i)}}(h).$$

The first term is a discrepancy term that measures how distinct different tasks are as measured by the probability of the disagreement of their individual hypotheses $h_{(i)}^*$ with that of $h_1^*$ under samples drawn from task $P_1$. We need to bound the second term on the right-hand side to prove Theorem 2. We have

$$\frac{1}{k} \sum_{i=1}^{k} \mathcal{E}_{P_{(i)}}^{1/\rho_{(i)}}(h) \leq \frac{1}{k} \sum_{i=1}^{k} \mathcal{E}_{P_{(i)}}^{1/\rho_{\max}}(h)$$

$$= \frac{1}{k} \sum_{i=1}^{k} \left( e_{P_i}(h) - e_{P_i}(h_i^*) \right)^{1/\rho_{\max}}$$

$$\leq \frac{1}{k} \sum_{i=1}^{k} e_{P_i}^{1/\rho_{\max}}(h) \leq e_{\bar{P}}^{1/\rho_{\max}}(h).$$

where the final step involves Jensen's inequality and $\bar{P} = 1/k \sum_{i=1}^{k} P_{(i)}$. This is the population risk of a hypothesis $h$ on the mixture distribution $\bar{P}$ and by uniform convergence, we can bound it as

$$e_{\bar{P}}^{1/\rho_{\max}}(h) \leq \left( e_{\bar{S}}(h) + c' \left( \frac{D - \log \delta}{km} \right)^{1/2} \right)^{1/\rho_{\max}}$$

for any $h \in H$, in particular $\hat{h}^k$, with probability $1 - \delta$. Putting it all together we have:

$$\mathcal{E}_{P_1}(h) \leq \frac{1}{k} \sum_{i=1}^{k} \mathcal{E}_{P_1}(h_{(i)}^*) + \frac{c}{k} \sum_{i=1}^{k} \mathcal{E}_{P_{(i)}}^{1/\rho_{(i)}}(h)$$

$$\leq \frac{1}{k} \sum_{i=1}^{k} \mathcal{E}_{P_1}(h_{(i)}^*) + \frac{c}{k} \left( e_{\bar{S}}(h) + c' \left( \frac{D - \log \delta}{km} \right)^{1/2} \right)^{1/\rho_{\max}}$$

$\square$

## D  FREQUENTLY ASKED QUESTIONS (FAQS)

1. **Why do you consider the setting with unlimited replay?**
   As mentioned in §6, we would like to ground the practice of continual learning. Our investigation is inspired by the existing work on continual learning and with this paper we seek to encourage future works to focus their investigations on key desiderata of continual learning, namely per-task accuracy and forward-backward transfer.

   With this goal, we are motivated by our results in Theorem 2 that fitting a single model on a set of tasks is fundamentally limiting in performance due to competition between tasks, this problem is only exacerbated by introducing the tasks sequentially. We have developed a general method named Model Zoo that, although designed for unlimited replay, can be executed in any of the standard continual learning settings. Our experiments show that Model Zoo significantly outperforms existing methods in all of these settings, including problem settings with no replay.

   We allow Model Zoo to revisit past data and grow its capacity iteratively in order to get to the heart of the problem of learning multiple tasks sequentially. In our view, if we can demonstrate effective continual learning without forgetting at least in this setting, it will provide a good foundation to build methods that conform to the stricter problem formulations.

   We believe that such a foundation is needed today if we are to advance the practice of continual learning. Let us explain why with an example. The simplest "baseline" algorithm named Isolated in our work, surprisingly outperforms all existing continual learning methods, without performing any data replay, or leveraging data from multiple tasks. An upper bound for performance of a continual learner is the accuracy obtained by a multi-task learner that has access to all tasks before training. We argue that a good continual learner's performance should lie in between the above two: it should be—at least–comparable to training the task in isolation, and as close to the performance of the multi-task learner as possible. The fact that existing methods perform much poorly than even Isolated indicates that we need to thoroughly investigate the tradeoffs that these methods make, e.g., while the single epoch setting helps mitigate forgetting, it has quite poor accuracy.

In short, we would like to argue that before we design new sophisticated methods for continual learning, we should take a step back and evaluate what simple methods can do and ascertain some level of baseline performance, so that we have a sound benchmark to compare the sophisticated method against. This is our rationale for considering the problem setting with unlimited replay. **We would also like to emphasize that Model Zoo is a legitimate continual learner because it gets access to each task sequentially, and has a fixed computational budget at each episode.** For a multi-task learner, the computational complexity scales with the number of tasks.

2. **Why do you call it continual learning, instead of, say, incremental or lifelong learning?**
   The current literature is quite inconclusive about the formal distinction between continual, incremental and lifelong learning. We have chosen to call our problem "continual learning" and, by that, we simply mean that the learner gets access to tasks sequentially instead of having access to all tasks before training begins.

3. **Why are you not using the same neural architectures as those in the existing literature? Perhaps the methods in this paper work better because you use a larger/different neural architecture.**
   We use a small deep network (WRN-16-4 with 3.6M weights) for all our experiments. In particular, this is smaller than the Resnet-12 or Resnet-18 architectures that are used in a number of continual learning experiments (see Kaushik et al. (2021)) and the Model Zoo has a comparable number of weights. The exceptional performance of Model Zoo indicates that these observations indicate that the significant gains in accuracy of Model Zoo are not simply a result of using a larger model. We also demonstrate results on continual learning with a much smaller model, a CNN with 0.12M weights (which entails that Model Zoo has about 2.42M weights). This is an extremely small model, and even this model, under all problem settings, improves the accuracy of continual learning over existing methods.

4. **Why not compare Model Zoo to ensemble versions of other methods?**
   We compare the performance of Model Zoo with ensemble versions of Isolated in Fig. 4. We observe that Model Zoo performs better than an ensemble of Isolated models. We did not compare against ensemble variants of existing continual learning methods because as our results show in multiple places, Isolated significantly outperforms the state of the art as a continual learner. We therefore expect that Model Zoo will also outperform ensembles of existing methods.

5. **Boosting is not novel.**
   We do not claim any novelty in developing boosting and moreover our method is only loosely inspired by it. The key property of Model Zoo that makes it effective is the ability to split the capacity of the learner across different sets of tasks, the ones that are chosen at each round. This entails that the implementation of Model Zoo is similar to that of boosting-based algorithms such as AdaBoost, but that is the extent of the similarity between the two. In particular, Model Zoo only uses the models that were trained on a particular task in order to make predictions for it. Unlike AdaBoost which combines all the weak-learners using specific weights, we simply average the predictions of all models trained on each task. To emphasize, boosting is not novel, but the ability of Model Zoo to split learning capacity across multiple models, one from each round, trained on a set of tasks, *is* novel.

6. **Identifying that tasks compete is not novel.**
   See §6 and the references in §2.1. The fact that tasks compete with each other is broadly appreciated–if not rigorously studied–in the theoretical machine learning literature. It is also appreciated broadly under the name of catastrophic forgetting in continual learning. Theorem 2 elucidates this competition and shows, together with Fig. 2, that it can be quite non-trivial. Even if some tasks compete, i.e., a hypothesis that is optimal for one performs poorly on the other, they may benefit each other if we have access to lots of samples from each task. An effective way to resolve this competition has been missing. Model Zoo is a simple and effective framework to tackle task competition; such a mechanism, and certainly its use for continual learning, is novel to our knowledge.

7. **Why does the rate of convergence in Theorem 2 depend upon $\rho_{\max}$, this seems quite inefficient.**
   The convergence rate in Theorem 2 which depends on $\rho_{\max}$ indeed seems pessimistic if one chooses a bad set of tasks to train together. But this may be a fundamental limitation of non-adaptive methods, e.g., that pool data from all tasks together to compute $\hat{h}^k$. If the

learner uses adaptive methods, e.g., if it has access to $\rho_{ij}$ and iteratively restricts the search space at iteration $k$ to only consider hypotheses that achieve a low empirical risk $\hat{e}_{S_{(i)}}$ on all tasks closer than $\rho_{(k)}$, then as (Hanneke and Kpotufe, 2020) shows, we can get better convergence rates if all tasks have the same optimal hypothesis. Let us note that we have chosen some drastic inequalities in Appendix C in order to elucidate the main point, and it may be possible to improve upon the rate.

8. **Can you give some intuition for the transfer exponent?**

The transfer exponent discussed in (5) is inspired by the work of Hanneke and Kpotufe (2020) and is defined by the smallest value such that

$$c\,\mathcal{E}_{P_i}^{1/\rho_{ij}}(h) \geq \mathcal{E}_{P_j}(h, h_i^*) = \mathcal{E}_{P_j}(h) + e_{P_j}(h_j^*) - e_{P_j}(h_i^*)$$

for all $h \in H$. This should be understood as a measure of similarity between tasks that incorporates properties of the hypothesis space. A small value of $\rho_{ij} \approx 1$ suggests that minimizing the excess risk on task $P_i$ (the left-hand side) is a good strategy if we want to minimize the excess risk on task $P_j$ (the right-hand side). But there may be instances when we can only reduce the left hand-side up to an additive term

$$e_{P_j}(h_j^*) - e_{P_j}(h_i^*)$$

that may be non-zero (or large) if the optimal hypotheses $h_j^*$ and $h_i^*$ perform very differently on samples from $P_j$. Mathematically, $\rho_{ij}$ is seen as the rate of convergence of the concentration term in Theorem 2 if samples from $P_i$ are used to select a hypothesis for $P_j$; larger the transfer exponent, more inefficient these samples, even if this additive term is zero.

