# OpenReview forum: "Model Zoo: A Growing Brain That Learns Continually"
_ICLR.cc/2022/Conference — ICLR 2022 Poster_

### Official Review · Reviewer_Cn5V · 2021-10-31

**Correctness:** 3
**Technical Novelty And Significance:** 3
**Empirical Novelty And Significance:** 3
**Recommendation:** 6
**Confidence:** 4

**Main Review:**

This paper provides both theoretical and empirical insights to support the claim that splitting the capacity of a model into several small learners can benefit continual learning, which have the potential to guide future continual learning research. While the empirical results to some extent corroborate the theory, the link between the outperformance of Model Zoo and the reasons predicted by the theory were not entirely obvious and could potentially be clarified with some additional experiments. Additionally, the experiments were independently lacking in some ways detailed below. I would consider increasing my score if these issues are largely addressed with revisions.

Comments:

- The theoretical analysis appears accurate to me and it demonstrates that if tasks are sufficiently related, then training them together with a shared feature extractor can benefit improve generalisation, but that if they are not, it can harm generalisation. The benefits of a shared feature extractor among some tasks is reflected in the design of ModelZoo, which trains a shared feature extractor along with task-specific classifiers for a subset of tasks at each round. Furthermore, the empirical results that the ensemble of isolated learners outperforms many existing methods and that ModelZoo sometimes outperforms the multi-task model provide evidence to the theoretical result that claims that it is not always beneficial to train all tasks in a single model. What was not entirely clear was why selecting tasks with high empirical risk under the ensemble is a good heuristic for selecting a subset that are synergistic in the way described in the theoretical analysis. Could it not be that, to paraphrase Tolstoy, each unhappy task is unhappy in its own way? The intuition here is was not clear to me. Two possible ways to empirically demonstrate that this method selects tasks that are more likely to be synergistic are:
    - Perform an analysis to show that groups of tasks selected by ModelZoo exhibit more transfer between each other than randomly selected groups of tasks.
    - Train an ablation to ModelZoo where the tasks are uniformly selected for training, rather than biased by empirical risk, and compare performance to model zoo.
- The experiments were lacking in a number of ways:
    - It is claimed that this paper is the first to perform continual learning on coarse cifar-100 (according to super-classes) but this does not appear to be true (see [1,2,3]).
    - In Table 1, no accuracy numbers are given for any of the competing methods on Split-CIFAR 10 nor for Coarse CIFAR-100, and only two methods are reported for Split MNIST, which is a very common benchmark in the literature. Split CIFAR-10, for example, was trained on in [7,8,9].
    - In Table 2, for the multi-epoch runs, no accuracy or forward/backward transfer metrics are reported for the competing methods, so it is difficult to compare them in terms of computation and storage to the methods proposed in the paper.
    - The definition of forward transfer is unconventional in that is simply the “accuracy on a new task when it is first seen”. Typically, as in [7], you would compare the accuracy on a task after having trained on each of the previous tasks compared to a baseline performance on a randomly initialised network.
    - None of the competing methods are reimplemented - all results are taken from the original papers or other ones that have reimplemented them. This is okay but it makes it difficult to compare them directly to the methods in the paper, e.g. to compare inference times, the models should ideally be trained on the same hardware or if not the differences should be clarified. Also, it is not clear if the training is equal across approaches in terms of how much replay data is afforded to each of them.
    - It would have been useful to empirically compare to at least one or two other continual learning methods that use ensembles (e.g. [4,5,6]) or at least to mention that other ensemble-based approaches to continual learning exist in the related work.

[1] Yoon, Jaehong, et al. "Scalable and Order-robust Continual Learning with Additive Parameter Decomposition." International Conference on Learning Representations. 2020.

[2] Yoon, Jaehong, et al. "Federated continual learning with weighted inter-client transfer." International Conference on Machine Learning. PMLR, 2021.

[3] Shanahan, Murray, Christos Kaplanis, and Jovana Mitrović. "Encoders and Ensembles for Task-Free Continual Learning." arXiv preprint arXiv:2105.13327 (2021).

[4] Rahaf Aljundi, Punarjay Chakravarty, and Tinne Tuytelaars. Expert gate: Lifelong learning with a network of experts. In Proceedings of the IEEE Conference on Computer Vision and Pattern Recognition (CVPR), pages 3366–3375, 2017.

[5] Yeming Wen, Dustin Tran, and Jimmy Ba. Batchensemble: an alternative approach to efficient ensemble and lifelong learning. In International Conference on Learning Representations (ICLR), 2020.

[6] Germán Kruszewski, Ionut-Teodor Sorodoc, and Tomas Mikolov. Evaluating online continual learning with calm. ArXiv preprint arXiv:2004.03340, 2021.

[7] Lopez-Paz David and Marc'Aurelio Ranzato. "Gradient Episodic Memory for Continual Learning". European Conference on Computer Vision. Advances in Neural Information Processing Systems. 2017.

[8] Rebuffi Sylvestre-Alvise, Alexander Kolesnikov and Christoph H. Lampert. "iCaRL: Incremental classifier and representation learning." arXiv preprint arXiv:1611.07725, 2016.

[9] Zenke, Friedemann, Ben Poole, and Surya Ganguli. "Continual learning through synaptic intelligence". International Conference on Machine Learning. 2017.


**Summary Of The Paper:**

This paper provides two main contributions on the topic of continual learning: (i) the first is to propose a definition of relatedness between tasks in the framework of statistical learning theory, and use it to perform a theoretical analysis of when it can be fruitful to train multiple tasks with the same model and when it can be detrimental to do so; (ii) the second is to propose an architecture for continual learning called ModelZoo, which maintains an ensemble of models that grows as each new task is introduced. Each time a new task arrives, a feature generator h and k task-specific classifiers g_k are initialised, and training proceeds on the combination of h and each g_k on the current task and data from k-1 previously seen tasks. At any point, the performance on a previously seen task can be evaluated by averaging over all small models previously trained on that task. ModelZoo takes inspiration from AdaBoost in how it selecting which previous tasks to train concurrently with the current one, which is by preferentially sampling tasks that have a bad performance under the current ensemble. An empirical analysis is performed by training on various standard continual learning image classification tasks, which demonstrate that Model Zoo is on par with and sometimes outperforms the single model multi-task approach (both , which is often used as an “upper bound” on continual learning methods. The results also demonstrate that a simple baseline of an ensemble of small isolated learners outperforms a selection of existing continual learning methods (with results reported reported from other papers).


**Summary Of The Review:**

This paper provides both theoretical and empirical insights to support the claim that splitting the capacity of a model into several small learners can benefit continual learning, which have the potential to guide future continual learning research. While the empirical results to some extent corroborate the theory, the link between the outperformance of Model Zoo and the reasons predicted by the theory were not entirely obvious and could potentially be clarified with some additional experiments. Additionally, the experiments were independently lacking in some ways detailed below. I would consider increasing my score if these issues are largely addressed with revisions.

---

> ### Author Response · Authors · 2021-11-18
> **Response to Reviewer Cn5V (1/2)**
>
> Thank you for your feedback.
>
> **>> What was not entirely clear was why selecting tasks with high empirical risk under the ensemble is a good heuristic for selecting a subset that are synergistic in the way described in the theoretical analysis. Could it not be that, to paraphrase Tolstoy, each unhappy task is unhappy in its own way? The intuition here is was not clear to me.**
>
> The theory predicts that capacity of the learner must be split among synergistic tasks. Selecting tasks with high empirical risk a pragmatic choice: it achieves capacity splitting but does not really guarantee the latter. Please also see the last comment to Reviewer LD4g (**“in the end it seems that the method does not use task synergies/competitions at all to decide which tasks to train on....)**
>
> This is however an interesting point. In general, some tasks may be more difficult than others and have high loss even if they are sampled in multiple episodes. A similar phenomenon is also seen in boosting algorithms where algorithms like AdaBoost can have poor performance for problems with noisy data whereby successive weak learners fixate on the noisy samples. We have not noticed such behavior in our experiments (see Appendix B.7 where some tasks have lower accuracies at the end than others but the differences are not usually drastic). In short, tasks can be unhappy in different ways (as Theorem 2 also shows) but usually for these datasets, there are enough shared traits that training such tasks together still improves their individual accuracy. This reason for this, perhaps, is the large learning capacity of deep networks, which is an interesting question for future work.
>
> **>> Perform an analysis to show that groups of tasks selected by ModelZoo exhibit more transfer between each other than randomly selected groups of tasks.**
> **or**
> **>> Train an ablation to ModelZoo where the tasks are uniformly selected for training, rather than biased by empirical risk, and compare performance to model zoo.**
>
> These are useful suggestions, thanks. The first one is quite difficult to do, but we did conduct the second experiment. The details of the experiment are present in section B.10.
>
> **>> It is claimed that this paper is the first to perform continual learning on coarse cifar-100 (according to super-classes) but this does not appear to be true (see [1,2,3]).**
>
> Thanks for pointing out these references, we were unaware of them. We will remove this sentence and cite these papers.
>
> **>>> In Table 1, no accuracy numbers are given for any of the competing methods on Split-CIFAR 10 nor for Coarse CIFAR-100, and only two methods are reported for Split MNIST, which is a very common benchmark in the literature. Split CIFAR-10, for example, was trained on in [7,8,9].**
>
> Table 1 in the main paper is was abridged to save space, Table A1 in the Appendix is its extended version and cites more methods. In particular, [7,9] are present in the extended version. We have added entries from [1, 2] to the table that compare to Coarse-CIFAR100 in the revised manuscript.
>
> [3] and [8] evaluate in the class-incremental setting, while our method focuses on the task incremental setting, due to which the numerical accuracies cannot be compared.
>
> We were unable to find competing methods for the task-incremental setting that evaluate on Split-CIFAR10. We believe [7] does not evaluate on Split-CIFAR10 or Split-MNIST. [9] evaluate on Split-MNIST but do not evaluate on Split-CIFAR10 (they instead evaluate on a datasets that mixes CIFAR10 and CIFAR100).
>
> **>>> In Table 2, for the multi-epoch runs, no accuracy or forward/backward transfer metrics are reported for the competing methods, so it is difficult to compare them in terms of computation and storage to the methods proposed in the paper.**
>
> A number of competing methods are designed for the single epoch setting and as noted in Appendix F of https://arxiv.org/pdf/1812.00420.pdf, their performance can be very poor if they are trained for multiple epochs. For Table 2, we therefore ran the competing methods (using code from the original authors) only in the single epoch setting that they were designed to be used and compute accuracy/forward-backward transfer. There are other methods in Table 1 and Table A1 that can be executed for multiple epochs but not all papers report continual learning curves such as those in Fig 1b or Appendix B.7 of our paper; this makes calculating these metrics quite difficult. To aid systematic comparison in future works, we provide extensive tabulated performance details of Model Zoo and also provide the learning curves for all tasks in continual learning.

---

> ### Author Response · Authors · 2021-11-18
> **Response to Reviewer Cn5V (2/2)**
>
> **>>> The definition of forward transfer is unconventional in that is simply the “accuracy on a new task when it is first seen”. Typically, as in [7], you would compare the accuracy on a task after having trained on each of the previous tasks compared to a baseline performance on a randomly initialised network.**
>
> This is a good point. Our definition of forward transfer is identical to the one in https://arxiv.org/pdf/1810.11910.pdf. We cite this in the manuscript now. Your suggestion to use the definition of forward transfer from [7] (which is akin to zero-shot transfer) is reasonable but it does not work for architectures with task-specific classifiers such as ours: the accuracy of the model trained on past tasks when evaluated on a new task using a new classifier head will essentially be identical to that of a randomly initialized network. Even in [7], their forward transfer evaluates to 0 for all methods on the Split-CIFAR100 dataset due to this.
>
> Our definition, which follows that of https://arxiv.org/pdf/1810.11910.pdf is a more useful measure of forward transfer. Roughly speaking, it characterizes the benefit of having seen past tasks to achieve a good accuracy on the new task as opposed to training the new task in isolation.
>
> **>>> all results are taken from the original papers or other ones that have reimplemented them. This is okay but it makes it difficult to compare them directly to the methods in the paper, e.g. to compare inference times, the models should ideally be trained on the same hardware or if not the differences should be clarified. Also, it is not clear if the training is equal across approaches in terms of how much replay data is afforded to each of them.**
>
> We have added the following text to the manuscript in Appendix A.5.
>
> In this section, we describe the methodology used to estimate training and inference times reported in Table 2.
>
> **Inference time.** The column titled inference time corresponds to per-sample prediction latency in milli-seconds. All entries for inference time in Table 2 were computed by us on an Nvidia V100 GPU and therefore they can be compared directly with each other. Note that inference times can be computed using only the architecture built by each method at the end of all continual learning episodes. We obtained the architectures used in each method from open-source implementations of the original authors (https://github.com/facebookresearch/agem and https://github.com/imirzadeh/stable- continual-learning). Inference time is computed by processing 50 mini-batches from CIFAR-100, each of batch-size 16. The inference time is computed by normalizing the total computation time by (size of mini-batch × number of mini-batches), which gives the average inference-time per sample. For Model-Zoo, we assume that inference time is approximately b = 5 times the inference time of isolated (where b is number of tasks sampled in every round).
>
> **Training time** corresponds to the time (in minutes) required to train all episodes of the Split- CIFAR100 dataset. Establishing an accurate comparison is difficult because different papers used different hardware but we have strived to be fair. The training time for EWC, Prog-NN, GEM and A-GEM are obtained from Chaudhry et al. (2019a) (we divide the numbers by 5 since this paper reports the sum of training times of 5 different runs). Chaudhry et al. (2019a) also report the training time for naive fine-tuning (21 mins) which in theory, should be very similar to the training time of our Isolated model (the training time for us is 20.76 mins on one V100 GPU). Since the two numbers are quite similar, we can estimate training time of the other continual learning methods using their computational cost relative to naive fine-tuning. Therefore, the estimate of the training times that we have reported in Table 2 can be compared to each other.
>
> **>>> It would have been useful to empirically compare to at least one or two other continual learning methods that use ensembles (e.g. [4,5,6])**
>
> Fig 4 (right) compares an ensemble of 100 Isolated models to Model-zoo and shows that naive ensembling does not yield the same performance as that of Model Zoo. Since our Isolated learner outperforms existing continual learning methods in Table 1, we can expect similar performance gaps between ensembles of existing continual learning methods and Model Zoo.
>
> Comparing to the specific ensemble methods cited by the reviewer [4,5,6] is difficult because these papers use different evaluation settings or datasets. [4] evaluates on MIT Scenes, Caltech-UCSB Birds and Oxford Flowers and not the more typical continual learning benchmarks considered in our paper. [5] evaluates on the class-incremental continual learning while we evaluate on the task-incremental continual learning. [6] evaluates exclusively on text datasets which are beyond the scope of this paper. Thanks for pointing out these references however, we will cite [4, 5] in our paper.

---

> > ### Comment · Reviewer_R6VS · 2021-11-22
> > **Question about assumption underlying inference time**
> >
> > Could I briefly ask why the assumption is made that the inference time for Model-Zoo is approximately b=5 times the inference time of Isolated? It seems straight-forward enough to me to measure the inference time for Model-Zoo as well.

---

> > > ### Author Response · Authors · 2021-11-23
> > > **Response: Inference Time**
> > >
> > > Model Zoo has a different number of models in the ensemble for different tasks. In these benchmark problems which typically have 20 episodes and b=5, we have observed that the ensemble typically has 5 models for most tasks. We therefore approximate the inference complexity as 5 times that of the Isolated (which is just 1 model/task). This is a generous over-estimate. Because Model Zoo has small models, a large number of them fit on the same GPU and we can parallelize inference in the ensemble easily.  We did not however do so because we wish to report inference complexity of the Zoo as a multiple of the inference complexity of the base model; this way the numbers in Table 2 can be used/scaled by future publications without having to rerun our experiments---just like we could do to estimate the training complexity of some past methods in Table 2.

---

### Official Review · Reviewer_KWuM · 2021-11-01

**Correctness:** 3
**Technical Novelty And Significance:** 2
**Empirical Novelty And Significance:** 2
**Recommendation:** 6
**Confidence:** 5

**Main Review:**

Pros:

The paper makes a very good case for the difficulties of sharing parameters between different tasks, which are amplified by sequential data access restrictions, especially without task labels. However, the paper doesn’t address many such issues, and simply goes on to assume many of these limitations don’t exist. That is fine, as long as the setup is clear from the start, and the tone of the paper reflects the reality of results, i.e. a simpler problem can be solved better than much harder problems.

The results provided show that even in favorable settings, e.g. splits of i.i.d. datasets, parameter sharing may not be optimal, even if data access restrictions are confined only to future data in sequential learning. This is an important contribution, but may not support all the claims.


Cons:

According to this paper, a continual learner’s only limitation is not knowing all data from all the tasks not yet seen; essentially, according to this definition, a continual learner’s only limitation is not knowing the future. Standard sequential learning settings used in the literature are borrowed and simplified by removing data access restrictions and giving full information about past tasks. Other implicit assumptions are made but not all are properly discussed:
- Data imbalance issues are not mentioned explicitly. What is the meaning of learning both MNIST and ImageNet?
- Task sequences considered are favorable to transfer and no attempt to study adversarial tasks is made;
- It is assumed past data can be neatly sorted into i.i.d. buckets (separate tasks) with known labels;
- Task losses aren’t always comparable in general, e.g. in terms of scale or learning dynamics; MNIST and CIFAR-10 have quite different error curves during learning; what is the meaning of the average joint objective?
In my humble opinion the paper should give a thorough discussion of limitations and appropriate qualifications for all claims.

Writing and framing issues:

The paper simply redefines continual learning in contradiction to a majority of recent literature. For example, the first paragraph of the paper gives a non-standard definition of continual learning: a continual learner isn’t widely defined as an algorithm which “leverages past data to learn new tasks”. While proposing a new definition is very interesting and indeed a potentially important contribution, the presentation of this new definition is at best misleading, significantly reducing the accuracy of the writing.

Many terms, e.g. “task relatedness”, task “competition for capacity”, “synergistic data” are offered without precise definitions or associated references. While difficulties in producing such precise definitions may be well understood, they don’t seem to temper the tone of the paper, nor do they seem to inspire the authors towards ideally a more humble, or at least balanced view of existing literature and the plurality of opinions within. The authors describe different viewpoints as an “intellectual gap” in research, which is not very helpful.

The authors appear to misattribute properties of the learning setting as qualities of the proposed algorithm, which I believe is done in error. The adopted CL setting offers data neatly split into labelled i.i.d. buckets. This does not mean that the data selection heuristic will always be effective, especially when these implicit assumptions are violated. However, the paper doesn't discuss such issues at length.

While the comments about focusing on backward and forward transfer are well taken, it is important to note that the proposed solution heavily relies on the definition of multi-task learning given originally by Caruana (1994), i.e. learning from multiple tasks independently as best suits each individual task. There is no guarantee that the proposed algorithm achieves this, however. Empirical results provided do not give good enough reasons to believe so because all experiments rely on splits of single datasets into well balanced bins with similar data, e.g. preprocessed in the same way, similarly centered and scaled, etc. Because the dataset shift is quite low between splits of X, it is really hard to tell how effective the proposed method is in the general case under its own assumed (much more permissive) CL setting. Indeed, at the beginning of section 3 the authors express hope that interference doesn’t happen, but offer no formal guarantee and do not take on an empirical challenge where the algorithm could demonstrate performance under less than ideal conditions.

It seems that the choice of how much data to use from tasks with higher loss is left as a hyper-parameter, and hence depends strongly on the data stream. If one can expect little interference, then many sources can be combined without much hindrance, as the paper observes; but this is not necessarily true in general. In fact, such hyper-parameters express the worst case expectations, which in general can be very negative. The paper does not discuss this, or report such results.


**Summary Of The Paper:**

This paper proposes yet another definition of continual learning (CL) and uses it to motivate a divide-and-rule approach. The algorithm exploits all assumptions of the adopted CL setting, including the relaxation of data access rules, in order to perform better compared to algorithms designed for more restrictive settings. The paper concludes that previous CL settings were holding back progress. Empirical evaluations and ablations are given using the new CL setting proposed in this paper, with the help of datasets favorable to this approach, namely splits of existing i.i.d. datasets.

**Summary Of The Review:**

This could be a great paper. It is important to understand what works in practice with current algorithms and architectures. Analysis of worst case performance is not the only important case in many practical settings. Hence, I lean towards acceptance despite the gross lack of nuance and generally imprecise writing. As I said, the authors could turn this paper into a great contribution by providing nuance instead of hyperbole, deference to the audacity of other researchers to face challenges which the authors avoid axiomatically, instead of off hand comments. I will (reluctantly) not oppose publication close to current form, but I cannot confidently recommend acceptance before the writing is improved.

---

> ### Author Response · Authors · 2021-11-18
> **Response to Reviewer KWuM (1/4)**
>
> Thank you for your feedback. We appreciate your comments and have addressed them in detail below. Your comments, regarding the writing, have significantly improved the quality of the paper. There are some comments (noted below) where we were a bit unsure of what you intend to convey, please let us know if our response satisfactorily addresses your concerns.
>
> **>>> The paper makes a very good case for the difficulties of sharing parameters between different tasks, which are amplified by sequential data access restrictions, especially without task labels. However, the paper doesn’t address many such issues, and simply goes on to assume many of these limitations don’t exist. That is fine, as long as the setup is clear from the start, and the tone of the paper reflects the reality of results, i.e. a simpler problem can be solved better than much harder problems.**
>
> We appreciate that there are a number of different settings for studying continual learning. Our work considers multiple settings as follows.
>
> 1. We investigate the task incremental setting where new tasks are shown to the learner in each episode. We consider a number of different sub-settings, e.g., when the learner is allowed to replay all past data (this is our standard setting), allowed to replay only a subset of the data (e.g., 10% of the samples) and where it does not replay any data (Isolated). We show results where each each datum is used exactly once (this is the single-epoch setting that a number of existing methods employ) and also settings where models are trained for multiple epochs.
> 2. Parameter sharing is difficult due to the sequential nature of the problem. Our theory in Section 2.2 identifies settings under which parameter sharing works, i.e., multiple tasks can be learned using a shared feature generator if their inputs (the same argument also holds for outputs) are simple transformations of each other. The theory in Section 2.3 shows settings where such sharing does not work, the first term in Theorem 2 is large if the same hypothesis is used for diverse tasks. Even experimentally, we show that parameter sharing, even in the non-continual setting, can sometimes perform worse than even the continual setting, see the entries in Table 1 where Model Zoo outperforms the multi-task learner.
>
> **Our goal is to study these different settings to understand continual learning across all of them.** **We do not seek to choose one specific setting and develop an algorithm for it.**
>
> **>>> , i.e. a simpler problem can be solved better than much harder problems**
>
> We do not completely understand what you mean by this comment.
>
> 1. Our data replay mechanisms do not make the problem easier or harder because we show results for multiple settings (see the previous comment).
> 2. If you instead mean that “not sharing parameters is easy and sharing parameters is hard”, we would argue that how the learner chooses to organize its architecture is a choice that it makes to achieve effective continual learning. Current methods lie on a broad spectrum of not sharing any parameters (e.g., our Isolated baseline, or Progressive Networks) to sharing all parameters and freezing/unfreezing capacity (RMN of Kaushik et al., 2021). Model Zoo employs a specific kind of parameter sharing, namely that of the model that is fitted on a set of tasks chosen at each episode.
> 3. If you intend to say that “known task identity is easier than unknown task identity”, we agree with you. But we would like to argue that studying the setting with known task identity is still meaningful for understanding continual learning. This is also evidenced by the large body of work that develops methods for continual learning with known task identities.

---

> > ### Comment · Reviewer_KWuM · 2021-11-27
> > **Thank you for the detailed rebuttal and constructively considering all suggestions!**
> >
> > I believe the writing has indeed improved; a more balanced view of the existing literature only adds to the quality of the paper. I still find the first paragraph misleading; while the proposed definition of continual learning is valid, it is not standard. It is an interesting definition, perhaps also relevant for some real-world settings, but why present it as if it was unanimously accepted. It is very well to argue for new definitions, and your paper can still be cited for proposing *this new definition* if you make it clear from the start that it is one among many. There are many cases where recording all the data is not feasible, e.g. video, or legal, e.g. for privacy reasons, or simply too costly to reuse all past data even if it could be recorded. Anyhow, that's just a small improvement I wanted to suggest!
> >
> >
> > I believe that putting brain in quotes is a good idea, glad you took that suggestion! I wanted to point out that some optogenetic evidence exists for "functional ensembles" learning and solving different tasks in rodent brains; while the brains themselves don't grow, different functional ensembles are assigned to different learned tasks, or different collections of tasks. Just as here, ensembles of different small subnetworks in the mouse brain are shown to work together to solve each task; the neuroscience result is quite strong, what is known as a double dissociation; we can make mice forget tasks selectively by (essentially) "undoing learning" in some subnetworks, but not others, equivalent here to deleting models in the zoo; synopsis: https://www.med.unc.edu/pharm/erasing-memories-of-mice-with-a-beam-of-light/ Article: https://www.nature.com/articles/nature15257. While this doesn't necessarily add much to your method, I believe a more apt metaphor is that of a *dynamic* "brain" that learns continually, rather than a *growing* "brain". I am quite sure that neuroscientists looking at your paper will balk at the current title because brains actually loose about a third of the neurons produced during early development. So they never actually grow significantly during learning per se, with tiny exceptions in the hippocampus. This is quite unfortunate, because the method may actually have some relevance beyond CL, but may be overlooked because of an unpalatable title. In contrast, the concept of a *dynamic* "brain" is quite popular, and I believe it is also more fitting to what's actually going on. Furthermore, it is also suggestive of future work, perhaps these ensembles could be further consolidated over time, as more data comes in? Some forgetting can be good forgetting if your collection of models is "refined" over time, perhaps according to some long term population *dynamics*, etc.
> >
> > Moving on to my other comments, I do appreciate the effort of authors to highlight that current CL benchmarks have issues, among other, admitting *surprising* solutions. This is a valid contribution. However, I believe these benchmarks were proposed with more constraints in mind, e.g. reduced past data access, little or no repetition of data, which means more learning is needed while data from a task is available, etc. In many cases "extreme" parameter sharing was also a goal. This is what I meant by taking a hard problem and making it easier. Each relaxation of the setting in isolation allows for different, more performant solutions. That's not surprising, imho, and is well supported by results in Figure 1. A good benchmark is supposed to be a stand-in for much harder problems, and to generally ranks algorithms in the same order as they would perform on the latter. Its constraints are designed in order to enforce that property. Hence, these constraints aren't meant to make sense for learning miniImageNet per se, and they don't, as your results well point out. But that's actually ok!
> >
> > Now, what I believe you are correct to point out, is that not all hard problems we care about are the same. In fact, data can be stored and continually used in learning for a number of them. That's how I would have started writing your paper. This is not even controversial, you only need to look at a few lifelong robotic learning papers. Essentially, you care about sufficiently different hard problems, and that's ok! Since the constraints of your favorite hard problems are different, it stands to reason that benchmarks for your setting are going to enforce different constraints. But is MNIST++/miniImageNet without strict data restrictions a good benchmark? I.e. will performance on these benchmarks tell us anything about the problems you really care about? I cannot find such proof in the paper, unfortunately. While some of the limitations are now well acknowledged in the text, I believe results on something like the visual decathlon challenge (https://www.robots.ox.ac.uk/~vgg/decathlon/) or CORe50 (https://vlomonaco.github.io/core50/) would have been more appropriate substrates, because they challenge your assumptions.

---

> > > ### Author Response · Authors · 2021-11-28
> > > **Response**
> > >
> > > Thank you so much for the elaborate and constructive feedback.
> > >
> > > **>>> I still find the first paragraph misleading; while the proposed definition of continual learning is valid, it is not standard. It is an interesting definition, perhaps also relevant for some real-world settings, but why present it as if it was unanimously accepted.**
> > >
> > > We would like to emphasize that we have not proposed a new definition of continual learning. The **goal/problem** of continual learning is to leverage past tasks to improve learning on future tasks and future tasks to improve learning of past tasks---all past work seeks to solve the same problem and there is nothing new that we say in this respect. As the reviewer agrees, there are many **formulations of this problem** with different motivations and relevance to different real-world settings. We would like you to **distinguish between the goal/problem and formulation of that problem.** We have not proposed a new problem. We have not propose a new formulation. We have simply shown results on a variety of existing problem formulations.
> > >
> > > **>> I believe a more apt metaphor is that of a dynamic "brain" that learns continually, rather than a growing "brain".... because brains actually loose about a third of the neurons produced during early development.**
> > >
> > > This is a great analogy and we are thinking along the exact same lines. The intellectual point that our theory makes here is that for an artificial learner to learn a sequence of tasks, the capacity has to grow with time due to competition between tasks. Unlike the biological brain, an artificial leaner can afford to do so. Neuroscience is not our field of research, so the following is is speculative, but we believe that Model Zoo-like ideas are analogous to learning in the brain after this early learning phase.
> > >
> > > Refining the models in the Model Zoo to ensure that the incremental capacity added after each episode goes to zero asymptotically is exactly how we are hoping to extend this work.
> > >
> > > **>>  However, I believe these benchmarks were proposed with more constraints in mind, e.g. reduced past data access, little or no repetition of data, which means more learning is needed while data from a task is available, etc. In many cases "extreme" parameter sharing was also a goal. This is what I meant by taking a hard problem and making it easier.**
> > >
> > > We agree with the reviewer. We have however shown in our paper that splitting the capacity of the continual learner is effective in each of the above constrained formulations, e.g., Isolated does not access past data, limited replay in the Model Zoo only replays a subset of the data, single epoch setting only access the data once, etc.
> > >
> > > We are not taking a hard problem and making it easier. We have shown that even in the hard settings, splitting the capacity of the learner is an effective strategy. Yes, we also shown the performance of the Zoo in the setting where infinite replay is allowed which the reviewer is calling “easy”. Regardless of whether this “easy” setting is relevant to real-world applications or not (see the next comment which identifies some settings where it might be relevant), for the purposes of this paper, it should be understood as a soft upper limit on the performance of the constrained continual learning formulations.
> > >
> > > The intellectual contribution of our paper is not just to show that relaxing these constraints improves performance, but rather to show that splitting capacity improves performance even in these constrained settings.
> > >
> > > **>> But is MNIST++ (CIFAR?) /miniImageNet without strict data restrictions a good benchmark? I.e. will performance on these benchmarks tell us anything about the problems you really care about?**
> > >
> > > A large body of existing work on continual learning uses MNIST, CIFAR- and ImageNet- based datasets. We have followed this literature and evaluated our method on these datasets. The formulation of Model Zoo is completely and can in principle tackle other datasets also; there are no “assumptions” in our formulation in Section 3.
> > >
> > > These datasets are certainly challenging enough benchmarks, as evidenced by the accuracy of existing methods on these problems. And they may also be practically relevant, e.g., AutoML-like systems like https://blog.google/technology/ai/introducing-pathways-next-generation-ai-architecture/ that tackle a large number of different tasks, or others like https://aws.amazon.com/rekognition/custom-labels-features/ that build custom models from data. There is a big difference between showing that a method works on benchmark problems and deploying it in the real world, and **until we actually attack a real-world problem, it is unclear whether the benchmarks are good, or whether they are not.** For some problems, e.g., autonomous navigation using computer vision datasets like CORe50 are great benchmarks, but for others like a cloud-based AutoML system, others like ImageNet or Visual Decathalon are better benchmarks.

---

> ### Author Response · Authors · 2021-11-18
> **Response to Reviewer KWuM (2/4)**
>
> **>>> According to this paper, a continual learner’s only limitation is not knowing all data from all the tasks not yet seen**
>
> **>>> Standard sequential learning settings used in the literature are borrowed and simplified by removing data access restrictions and giving full information about past tasks.**
>
> **>>> However, the paper doesn’t address many such issues, and simply goes on to assume many of these limitations don’t exist.**
>
> We do not make the claim that a “a continual learner’s only limitation is not knowing all data from all the tasks not yet seen”. The *goal* of continual learning is to leverage data from past tasks to improve learning on future tasks and that of future tasks to improve learning of past tasks. There are many hurdles (limitations) to achieving this goal, e.g., that the tasks are shown sequentially and parameters need to shared (to achieve statistical efficiency) but also capacity also needs to be split (to mitigate competition). Different methods in the literature focus on different limitations, but with the same goal.
>
> About your comment on sequential learning, please see our response to your first comment. We do not simplify the problem to remove data restrictions. Yes, we do show how effective data replay can be, but we also show that Model Zoo outperforms existing methods without data replay, and also with limited data replay.
>
> We emphasize that we do not assume that these limitations do not exist. We study the problem under a number of different limitations (the updated narrative makes this methodology explicit).
>
> **>>> Implicit assumptions: Data imbalance issues are not mentioned explicitly. What is the meaning of learning both MNIST and ImageNet?**
>
> Model Zoo is agnostic to the number of samples in different tasks because it uses the average loss across a task for both training and evaluating the boosting weights. All continual learning benchmark problems in our paper come from existing literature. All problems have been designed such that each task has an equal number of samples (see Appendix A.1 for the configuration of each problem). Do note that the columns corresponding to MNIST, CIFAR-100, ImageNet in Table 1 are entirely different experiments, i.e., we do not do experiments where a few tasks come from MNIST and a few other tasks come from ImageNet (this is also the case with most existing methods in the literature).
>
> **>>> Task sequences considered are favorable to transfer and no attempt to study adversarial tasks is made**
>
> We use splits that are widely used in existing work. But this is an interesting question. It is unclear whether the task sequences considered in our work (and existing literature) are favorable or unfavorable to continual learning, e.g., even estimating distances between tasks for transfer learning is quite difficult (e.g., see https://arxiv.org/abs/1902.03545 or https://arxiv.org/abs/2011.00613 and references therein). To emphasize, we do not claim/attempt to study adversarial task sequences.
>
> **>>> It is assumed past data can be neatly sorted into i.i.d. buckets (separate tasks) with known labels**
>
> Yes, we have now made this clear in the remark in Section 3.
>
> **>>> Task losses aren’t always comparable in general, e.g. in terms of scale or learning dynamics; MNIST and CIFAR-10 have quite different error curves during learning; what is the meaning of the average joint objective? In my humble opinion the paper should give a thorough discussion of limitations and appropriate qualifications for all claims.**
>
> Let us first emphasize that we do not mix tasks from MNIST and CIFAR. Each continual learning problem uses tasks created from the same dataset (see Appendix A.1) and each task also has the same number of output classes. Strictly speaking, the loss of a task is the average cross-entropy loss across its samples and a surrogate for the zero-one error, so it can be compared across tasks. Yes, the dynamics of learning can be different (some tasks can be fitted with fewer epochs than others). The Model Zoo does not depend/rely upon the dynamics of the tasks. We show that even without modeling the learning dynamics we can still significantly outperform existing continual learning methods. This is decidedly a merit of our method, not a limitation.

---

> ### Author Response · Authors · 2021-11-18
> **Response to Reviewer KWuM (3/4)**
>
> **>>> The paper simply redefines continual learning in contradiction to a majority of recent literature. For example, the first paragraph of the paper gives a non-standard definition of continual learning: a continual learner isn’t widely defined as an algorithm which “leverages past data to learn new tasks”. While proposing a new definition is very interesting and indeed a potentially important contribution, the presentation of this new definition is at best misleading, significantly reducing the accuracy of the writing.**
>
> Respectfully, we do not provide a new definition of continual learning. Leveraging past data to learn new tasks is the “goal” of continual learning. This is of course not a contradiction with a majority of literature, both old and new, as evidenced by the focus on forgetting, forward/backward transfer etc. Broadly speaking, we wish to delineate the goal of continual learning from the approach used to achieve continual learning. While the goal is to learn new tasks efficiently, the approaches are indeed diverse, e.g., all shared parameters, no replay, unknown task identities, infinite replay etc. Let us emphasize that we do not consider non-standard approaches (also see the first comment in the response).
>
> **>>> Many terms, e.g. “task relatedness”, task “competition for capacity”, “synergistic data” are offered without precise definitions or associated references**
>
> Section 2.2 defines one notion of task relatedness through the family of functions F. Section 2.3 defines another notion of relatedness using the transfer exponent in equation (5). The competition term (the first term on the right-hand side in equation (6)) is explained in the narrative below Theorem 2 and in Remark 3. Synergistic tasks are explained in Remark 4, they are tasks with a small value of transfer exponent.
>
> **>>> While difficulties in producing such precise definitions may be well understood, they don’t seem to temper the tone of the paper, nor do they seem to inspire the authors towards ideally a more humble, or at least balanced view of existing literature and the plurality of opinions within.**
>
> Please see the previous comment, we do define these quantities. We appreciate the plurality of opinions and formulations of continual learning and have modified the narrative to reflect this. Please see the common response to all reviewers and the annotations in blue in the PDF.
>
> **>>> The authors describe different viewpoints as an “intellectual gap” in research, which is not very helpful.**
>
> We appreciate this comment. We have removed this phrase from the paper (it was in the last paragraph in Section 1). Please note that we do not wish to deride different viewpoints but merely wanted to point out the surprising fact that an Isolated learner performs so well at continual learning on benchmark problems.
>
> **>>> The adopted CL setting offers data neatly split into labelled i.i.d. buckets. This does not mean that the data selection heuristic will always be effective, especially when these implicit assumptions are violated.**
>
> You are right, this point is related to your earlier point about assuming that the task identity is known. Our method (like many other methods in the literature) assumes that the task identity is known for every datum that is presented to the continual learner. The data selection heuristic also exploits this knowledge.
>
> If this assumption is violated, the learner needs some mechanism to guess the task identity, e.g., as done by Kaushik et al., 2021 in RMN. We believe that we can incorporate such ideas in Model Zoo but that is beyond the scope of the present paper.
>
> **>>> It is important to note that the proposed solution heavily relies on the definition of multi-task learning given originally by Caruana (1994), i.e. learning from multiple tasks independently as best suits each individual task. There is no guarantee that the proposed algorithm achieves this, however.**
>
> Yes, our work is inspired by Caruana work and a lot of theoretical machine learning work that followed it, e.g., see the references to a number of papers by Shai Ben-David and co-authors. We agree that there is no guarantee that Model Zoo identifies the correct set of synergistic tasks to train with. In fact, as we explain in Section 3, it is not designed to do so. Model Zoo is an algorithm inspired by our theory and although it does not implement the theory exactly, it implements the crux of its implications, namely that the learning capacity should be split among sets of tasks. Please see the comment to Reviewer LD4g (**“in the end it seems that the method does not use task synergies/competitions at all to decide which tasks to train on....)**

---

> ### Author Response · Authors · 2021-11-18
> **Response to Reviewer KWuM (4/4)**
>
> **>>> Empirical results provided do not give good enough reasons to believe so because all experiments rely on splits of single datasets into well balanced bins with similar data, e.g. preprocessed in the same way, similarly centered and scaled, etc. Because the dataset shift is quite low between splits of X, it is really hard to tell how effective the proposed method is in the general case under its own assumed (much more permissive) CL setting. Indeed, at the beginning of section 3 the authors express hope that interference doesn’t happen, but offer no formal guarantee and do not take on an empirical challenge where the algorithm could demonstrate performance under less than ideal conditions.**
>
> The points you have raised here are well taken. We would however like to emphasize some important details.
>
> 1. The splits we consider are the same as the ones in a large number of existing papers on continual learning. We do not construct unduly simplistic problems. So at least the comparison with other methods is fair. For example, as you say, combining data from different datasets will potentially create more complex continual learning setups and evaluating Model Zoo in such (more realistic) settings is a very interesting prospect. We believe that the simplicity of the continual learning mechanism in Model Zoo makes it less susceptible to changes in image size/statistics/domain etc. But we do not tackle this setting.
> 2. Tasks need not always come from different domains for continual learning to be challenging. Our paper provides a number of examples. Fig 2 shows how even for CIFAR-100, accuracy on one task can vary a lot depending upon which tasks it is co-trained with, even in the multi-task setting; continual learning where the learner may not get synergistic sets of tasks (e.g., the ones with green cells in Fig 2) only makes things harder. A similar result is also seen in Fig A1 where we show the entire matrix of pairwise dissimilarities between tasks from CIFAR-100. So we disagree with the reviewer when they say that “dataset shift is quite low between splits”---if this were the case, then continual learning for existing  benchmark problems would be easy and this would also have likely been reflected in the accuracy of the existing methods.
>
> **>>> do not take on an empirical challenge where the algorithm could demonstrate performance under less than ideal conditions**
>
> We appreciate your comment. Now that we have a method that has good performance on existing problems, we would love our future work to tackle more challenging continual learning setups. This is beyond the scope of the present paper.
>
> **>>>  It seems that the choice of how much data to use from tasks with higher loss is left as a hyper-parameter, and hence depends strongly on the data stream.  If one can expect little interference, then many sources can be combined without much hindrance, as the paper observes; but this is not necessarily true in general. In fact, such hyper-parameters express the worst case expectations, which in general can be very negative. The paper does not discuss this, or report such results.**
>
> The number of samples to use from past tasks while fitting a model at each episode is not a hyper-parameter. But perhaps we do not fully understand what you exactly mean by “how much data to use from tasks with higher loss”. We clarify with an example.
>
> If Model Zoo uses b = 3, at each episode, it samples two of the past tasks in addition to the current task. In the setting where unlimited replay is allowed, it uses all data from all three tasks and fits the model for multiple epochs (200 epochs). In the limited replay setting, say 10% replay, Model Zoo samples mini-batches from a fixed subset of 10% of the samples of the two past tasks and all samples of the current task, this model is also trained for multiple epochs (200 epochs of the current task). The mini-batch size is fixed to 16 for all our experiments (see Appendix A.3 which discusses how it was tuned). Using the single epoch setting where the current task is trained only for one epoch (as is commonly done in existing methods in Table 1 and Table 2), we have conducted ablation studies on both the fraction of replay allowed to the learner (Fig 4 left) and the number of past tasks sampled at each episode (Fig 4 middle). In both cases, we see that the performance of Model Zoo is consistent.
>
> To summarize, we do not think our results are sensitive to hyper-parameters because all hyper-parameters are fixed for all experiments (across problem settings, datasets, and architectures).

---

### Official Review · Reviewer_LD4g · 2021-11-02

**Correctness:** 4
**Technical Novelty And Significance:** 3
**Empirical Novelty And Significance:** 3
**Recommendation:** 8
**Confidence:** 3

**Main Review:**

I am not an expert in continual learning, so there are details that I might have overlooked. Overall I think this is a strong paper.

One of the confusions I had as a reader: The authors talk a lot about task synergy and competition (e.g., sections 2.2 and 2.3, theorem 2, figure 3, etc.) However, in the end it seems that the method does not use task synergies/competitions at all to decide which tasks to train on. Instead, tasks are more likely to be chosen if they have high loss under the existing model (equation 8). I would appreciate a bit more discussion about this. Is the current method a proxy for selecting synergistic tasks? If not, how is the preceding discussion relevant?

**Summary Of The Paper:**

This paper begins with a clear exposition of continual learning and an analysis from a statistical learning theory perspective.

It then introduces the Model Zoo: A sequence of small models that are each trained on a subset of the available tasks at that point (with a separate linear classifier for each task). The tasks selected for training are those that have high loss under the models trained so far (similar to boosting).

The following section contains an exhaustive set of experiments that compares the Model Zoo with a variety of baselines (with and without replay, with and without limited training, a single model with multiple heads, etc.). Surprisingly, the "Isolated" model (i.e., a separate model is trained on each task) does better than most baselines that train on multiple tasks. The Model Zoo outperforms all other baselines and even outperforms the multi-head baseline on ImageNet.

The paper ends with a critical discussion on how the problem of continual learning should be approached.

**Summary Of The Review:**

I think this is an excellent paper. It is well-written, the proposed algorithm has strong results, it asks important and critical questions about continual learning research, it validates the proposed model with a large amount of experiments, is well-referenced, and provides source code and 14 pages of appendices to ensure reproducibility of the results.

---

> ### Author Response · Authors · 2021-11-18
> **Response to Reviewer LD4g**
>
> Thank you for your feedback. We are glad that you think our work is important for continual learning research and appreciate your review of material in the Appendix.
>
> **>>> In the end it seems that the method does not use task synergies/competitions at all to decide which tasks to train on. Instead, tasks are more likely to be chosen if they have high loss under the existing model (equation 8). I would appreciate a bit more discussion about this. Is the current method a proxy for selecting synergistic tasks? If not, how is the preceding discussion relevant?**
>
> Section 2 justifies why splitting capacity is beneficial to obtain good generalization on each task. This is relevant to continual learning because a large number of existing methods focus on different ways to update one model to handle new tasks. The theory in Section 2 identifies one situation when such an approach is likely to work well, namely when tasks are related to each other via simple transformations of inputs or outputs (Section 2.2), and also identifies situations when such an approach does not work, namely when tasks compete with each other (Section 2.3).
>
> Model Zoo is directly inspired by this theory. It is however difficult to implement the theory exactly. For instance, this would require estimating the transfer exponent in (equation 5) to identify synergistic tasks (see Fig. A1 in the Appendix which can be understood as an estimate of the transfer exponent). Theorem 2 also identifies the ideal way to split capacity (by arranging tasks in order of increasing transfer exponent). We cannot identify synergistic tasks easily, especially in the continual setting. But the crux of the theory is that the learner’s capacity should be split across sets of tasks. This idea can be implemented using heuristics. In this work, we use the training loss to select tasks that could benefit from further training. As you mention, this does not mean that such tasks are synergistic. Let us note that Model Zoo can be extended in a number of ways to better capture all aspects of the theory, e.g., by using second models to estimate the transfer exponent.

---

### Official Review · Reviewer_R6VS · 2021-11-03

**Correctness:** 2
**Technical Novelty And Significance:** 4
**Empirical Novelty And Significance:** 4
**Recommendation:** 5
**Confidence:** 4

**Main Review:**

Strengths:
-	The proposed boosting-style algorithm for task-incremental learning is an original and important contribution to the continual learning literature.
-	The surprisingly large gap in performance between the Isolated learner and the compared existing methods is intriguing.
-	The theoretical analyses and considerations in Section 2 are interesting, and I think they provide important insights and food-for-thought for many continual learning researchers.

As stated above, I believe this paper makes important and insightful contributions, but there are several issues that should be addressed before this paper could be published at this conference:
-	The paper suggests that it covers virtually all continual learning settings out there, but the paper ignores several important settings. Specifically, a critical assumption that this paper makes for all its experiments is that task identity is always provided to the learner (i.e. task-incremental learning). The paper does not consider settings where task identity is not provided (i.e. domain- and class-incremental learning; https://arxiv.org/abs/1904.07734). In a footnote it is argued that without the assumption of task identity being provided continual learning is impossible in the case of permutation of classes; but it is unclear why this argument is relevant as none of the experiments considered involve such a permutation of classes. It is important that it is made clear that the insights and observations of this paper are specific to the task-incremental learning setting.
-	At several places, the paper makes claims about performing better than all existing methods, on all datasets and on all benchmarks. Based on the reported experiments, these claims are too strong (e.g. it is clear not all existing methods are compared against), and I also think that these claims are unlikely to be true (e.g. see below). These claims should therefore be moderated.
-	Perhaps most importantly, I have serious concerns about whether the comparisons against existing continual learning methods (e.g. in Fig 1A, Table 1) are done fairly. Firstly, for some results it is stated that they are taken from other publications. This seems problematic, as the mentioned publications use different architectures and optimization methods. It is therefore unclear whether the reported difference is due to these differences (which are arguable unrelated to continual learning), or due to the continual learning strategy. Secondly, I wonder how the other results (i.e. those for which it is not stated that they are taken from other publications) were obtained. I could not find any description of this in the paper, and also in the submitted code there does not seem to be any code for these other methods.
I am particularly worried about this because from Fig 1A, it seems to be the case that the main part of the reported improvement over existing methods is due to an improved ability to learn individual task (e.g. compare all accuracies after just the first), rather than due to an improvement in forward/backward transfer. That is, I am afraid that the main improvements reported in this paper might be due to network architecture/optimization choices that are not interesting from a continual learning perspective.
-	An important class of methods that is not considered in this paper is generative replay (Mocanu et al., 2016 https://arxiv.org/abs/1610.05555; Shin et al., 2017 https://arxiv.org/abs/1705.08690), which does not store data and could be interpreted as adhering to the “strict formulation” of continual learning. I think it is important to at least discuss these methods as I am positive that some more recent generative replay variants (e.g. van de Ven et al., 2020 https://www.nature.com/articles/s41467-020-17866-2; Liu et al., 2020 https://arxiv.org/abs/2004.09199) could outperform the Isolated baseline.
-	It is claimed that the “Multi-Head (multi-task)” baseline is not a continual learner because it is jointly trained on all tasks. However, it seems straight-forward to adapt this baseline in such a way that it is, for the purposes of this paper, a “legitimate continual learner”: the tasks are visited sequentially as in the other baselines, and on each new task, the model is trained using the data of all tasks seen so far. (Or in other words, always all previous tasks are replayed fully.)
-	It is unclear to me how Model Zoo with limited replay is performed, and it would be good to provide some more details. In the text it is stated that only a subset of the data is used in equation (7), but I think the authors probably meant to refer to another equation?
-	The claim in the title that the proposed Model Zoo is a “Growing Brain” does not appear to be supported or justified anywhere in the text. I think it would be good to either modify the title, or to justify and discuss the reasons for this comparison in the text.


**Summary Of The Paper:**

This paper argues and demonstrates that in a task-incremental continual learning setting, in many cases it is beneficial to split up the capacity of a learning algorithm and to learn separate models for different tasks. Building on this insight, and taking inspiration from the boosting literature, the paper then proposes the method Model Zoo, in which a new model is learned for each new task and whereby each new model is trained on multiple tasks. During evaluation, the predictions of all models trained on the task under consideration are averaged. The paper reports surprisingly large improvements over existing continual learning methods on several different task-incremental learning methods, both with and without utilizing replay.

**Summary Of The Review:**

Although I believe this paper provides several important insights, there are several critical issues that prevent me from supporting acceptance of this paper. I am happy to engage in a discussion with the authors, to see whether my concerns can be taken away or whether the paper could be improved.

---

> ### Author Response · Authors · 2021-11-18
> **Response to Reviewer R6VS (1/3)**
>
> Thank you for your comments and willingness to engage in a discussion. We are glad that you find the insights in this manuscript important and appreciate the points that you have raised in the review. We have modified the manuscript at various places (in blue) to address these concerns. We enclose our replies below and look forward to the discussion.
>
> **>>> Assumes that the task identity is known at test time (this is the task incremental setting)... It is important that it is made clear that the insights and observations of this paper are specific to the task-incremental learning setting.**
>
> We agree with the reviewer, our paper only considers settings where the task identity is known at test time. We have added a remark that makes this clear in Section 3.
>
> **>>> In a footnote it is argued that without the assumption of task identity being provided continual learning is impossible in the case of permutation of classes; but it is unclear why this argument is relevant**
>
> The footnote has been removed from the revised version of the manuscript.
>
> **>>> At several places, the paper makes claims about performing better than all existing methods, on all datasets and on all benchmarks. Based on the reported experiments, these claims are too strong. These claims should be moderated.**
>
>  We have moderated the narrative at various places (see the text in blue) to remove words like “all settings” and “all datasets” and made clear what the exact settings (different existing variants of task incremental settings) and datasets are.
>
> **>>> Are the comparisons in Fig. 1a and Table 1 fair?**
>
> **>>> Firstly, competing methods use different architectures. I am afraid that the main improvements reported in this paper might be due to network architecture/optimization choices that are not interesting from a continual learning perspective.**
>
> A number of experiments in the paper indicate that our performance is not due to the choice of a specific architecture, hyper-parameters or optimization method. Let us explain.
>
> **Hyper-parameters and Optimization Methods** All hyper-parameters are fixed for all datasets and all experiments in our paper. Our training methodology is quite standard (SGD with Nesterov’s momentum).
>
> **Architecture** We use rather small networks in our paper (the WRN16-4 architecture which is used in Model Zoo has 3.6M weights and the CNN which corresponds to its “small” version has 0.12M weights without any residual connections). Methods in literature have experimented with larger networks (Resnet-12, Resnet-18) or their smaller variants (Renset-18-S with 1.6M weights) but both choices lead to poorer accuracy than our method (both the WRN model with 3.6M weights and the CNN with 0.12M weights). The number of parameters of the small version of Model Zoo is comparable to these architectures (see Table 2). Both networks achieve similar performance in Table 1 that is significantly better than existing methods. This indicates that the performance does not come from the choice of the specific architecture. If a three-layer CNN which is much simpler than complex modern architectures also performs well under Model Zoo (and outperforms existing methods), the performance of the Zoo likely comes from the algorithm and not the architecture.
>
> To elaborate upon this, in the single epoch setting (Fig 1a), the architecture does have an influence on performance. For example, Model Zoo-small (dark blue, 81%)  is better than Model Zoo (cyan, 64%), and the trend is the same for Isolated small vs. standard. This is because all these models are severely undertrained in the single epoch setting. For the multi-epoch case, both the small and standard version perform very well (see Fig 1a and Table 1). This indicates that the practice in the existing literature of training for a single epoch makes the conclusions sensitive to the architecture. One should note that competing methods suffer a drastic reduction in performance in the multi-epoch setting (this is also mentioned in A-GEM, Chaudhary et al., 2019a).
>
> **>>> Secondly, I wonder how the other results (i.e. those for which it is not stated that they are taken from other publications) were obtained**
>
> Numbers on Split-MiniImagenet/Coarse-CIFAR100 were obtained using open-source implementations by the original authors (https://github.com/facebookresearch/agem, https://github.com/imirzadeh/stable-continual-learning) because these methods were not evaluated on these two datasets in the original publications. We have also since found a paper (we will cite this), e.g., [https://arxiv.org/abs/2105.05155](https://arxiv.org/pdf/2105.05155.pdf), which reports similar numbers to ours for these methods. Entries for all other datasets in Table 1 and Table 2 are taken from the numbers reported in the respective publications.

---

> > ### Comment · Reviewer_R6VS · 2021-11-19
> > **Main concern not taken away**
> >
> > Thank you for your extensive response and for clearly indicating the changes in the paper. This is much appreciated.
> >
> > I still need to go through most of your responses in more detail, but given the tight timeline, I wanted to already get back to you about the most pressing issue.
> >
> > I’m afraid that my main worry has not been taken away: that a large part of the reported improvements over existing methods could be due to network architecture/optimization choices rather than due to algorithmic improvements that are relevant from a continual learning point of view.
> >
> > My understanding is that the Model Zoo methods are run with a WRN16-4 architecture (either a large or a small version), while the competing methods (e.g., EWC, AGEM, ER) were run with different architectures (Resnet-12, Resnet-18 or Resnet-18-S). In the rebuttal, the authors argue that the performance improvement they report is not due to the choice of architecture because the small version of their WRN16-4 architecture has a comparable number of parameters to the architectures used by other methods. This argument does not hold. There can be substantial differences in performance between two architectures with comparable number of parameters.
> > I’m afraid that therefore, based on the presented experiments, it cannot be ruled out that the reported performance improvements *might* be due to these differences in architecture.
> >
> > But I think the problem is more severe than this, because based on Figure 1 (Left), it even seems *likely* that a large part of the reported improvements are due to these differences in architecture. Let’s look at the performance of all compared methods after just the first task. Because at that point only one task has been learned, there hasn’t been any “continual learning aspect" yet, so it seems fair that any differences between the methods can be attributed to aspects that are not (necessarily) interesting from a continual learning perspective (e.g., differences in architecture). Strikingly, after the first task, the performance of the Model Zoo methods (~90% for large and ~60% for small) is substantially higher than the performance of the existing methods that are compared against (35-50%). These differences after the first task seem to explain most, if not all, of the reported performance improvements after all tasks have been learned.
> >
> > In short, I’m afraid that it is possible – and probably likely – that most of the reported performance improvements in this paper are due to differences in network architecture (or perhaps optimization), and that it is therefore unclear what the significance of these results is from a continual learning perspective.

---

> > > ### Author Response · Authors · 2021-11-20
> > > **Response (1/2)**
> > >
> > > Thank you for engaging in a discussion. We believe the reviewer has misunderstood some of the experiments, which we have clarified below. We also conducted an additional experiment to address this concern thoroughly. We explain these below. We hope this response addresses the concerns of the reviewer and encourage them to get back to us if it does not.
> > >
> > > **>> My understanding is that the Model Zoo methods are run with a WRN16-4 architecture (either a large or a small version),**
> > >
> > > No. Model Zoo is run with two architectures, (i) a WRN-16-4 with 3.6M weights, and (ii) **a vanilla CNN with 3 layers and 0.12M weights in total.** **To emphasize, we are **not** simply using two variants of the WRN-16-4 architecture; these are two different architectures.** The latter is an extremely small network; even Lenet-5 that is used for standard supervised learning on MNIST has more weights (0.13M).
> > >
> > > Model Zoo-small using this three layer CNN has 92.06% average accuracy in Tabl 1 on Split-CIFAR100 and 73.72% accuracy on Coarse-CIFAR100; these are both more than 10% better than existing continual learning methods which use architectures that are larger and more complicated (residual networks with variants). If such a small network without any bells and whistles performs so well, surely the performance is coming from the method and not the architecture.
> > >
> > > **In fact, even Isolated-small outperforms existing methods.** **This is even more surprising.** **This suggests that such a small network, if trained correctly up to convergence, is effective.** As we explained in the previous response, we do not do anything special for training. We simply train all networks till convergence (as they should be trained for any problem).
> > >
> > > **>> Additional experiment**
> > >
> > > To further address this concern, we conducted a new experiment that evaluates Model-Zoo and Isolated on the Split-CIFAR100 dataset using the Resnet18-S architecture (used in works like A-GEM,GEM,ER). We use the architecture implementation provided by the original authors of GEM (https://github.com/facebookresearch/GradientEpisodicMemory/blob/master/model/common.py).
> > >
> > > **We observe the same trends for the Resnet18-S architecture as those in our original paper, i.e. Model Zoo does better than Isolated, and Model Zoo/Isolated both outperform competing methods in both the single/multi-epoch settings.**
> > >
> > > **Both Model Zoo/Isolated have similar accuracies on Split-CIFAR100 with 3 different architectures and all of them are better than existing methods. This is conclusive evidence that the improved accuracy is not a result of the choice of a specific architecture.**
> > >
> > > We have added these entries to Table 1 in the manuscript.
> > >
> > > |	|Multi-epoch Accuracy (%)	|Single-epoch Accuracy (%)	|
> > > |---	|---	|---	|
> > > |Isolated (small)	|90.18	|71.6	|
> > > |Model Zoo (small)	|92.06	|73.67	|
> > > |Isolated	|91.9	|50.43	|
> > > |Model Zoo	|94.99	|57.67	|
> > > |Isolated (Resnet-18-S)	|88.95	|65.41	|
> > > |Model Zoo (Resnet-18-S)	|93.15	|69.3	|
> > > |A-GEM	|- 	|62.3	|
> > > |RMN	|80.01	|-	|
> > >
> > > As expected for a larger architecture, Model Zoo (Resnet-18-S) has a slightly larger accuracy than Model Zoo (small); Model Zoo (Resnet-18-S) has 32M weights while Model Zoo (small) has 2.4M weighs.
> > >
> > > **>> Another evidence as seen from Table 2 of the manuscript**
> > >
> > > The following table is an excerpt from the single-epoch columns of Table 2 of the manuscript.
> > >
> > > |	|Average Forward transfer (%)	|
> > > |---	|---	|
> > > |EWC (Resnet18-S)	|67.76	|
> > > |GEM (Resnet18-S)	|67.61	|
> > > |A-GEM (Resnet18-S)	|70.13	|
> > > |Isolated (Small)	|71.6	|
> > > |Isolated	|50.43	|
> > >
> > > Forward transfer (as defined in Section 4.2) refers to the accuracy of the task after it is first seen.  **This table shows that different methods learn the task to a similar accuracy when it is first seen. In fact the last row with Isolated (WRN16-4) is far worse than other architectures in the single epoch setting. And yet, Model Zoo (WRN16-4) which uses this same base model for continual learning outperforms existing methods (cyan line in Fig 1a).** This directly shows that the performance of Model Zoo is not coming from simply picking a different architecture.

---

> > > > ### Comment · Reviewer_R6VS · 2021-11-22
> > > > **Concerns about the fairness of the reported comparisons**
> > > >
> > > > Thank you for your responses and clarifications. I had indeed misinterpreted some of the reported experiments, my apologies. The additional explanations and experiments provided by the authors take away some of my earlier concerns, but I’m afraid they also raise new ones.
> > > >
> > > > In particular, I had not appreciated the extent to which the authors compare against methods trained only for a single epoch per task. Between the main results presented in Fig 1 and Table 1, most (or all, in case of Fig 1) of the compared methods are trained with a single epoch, while most (or all, in case of Table 1) of the Isolated/ModelZoo are trained for multiple epochs. It is clear that this is not a fair comparison: on the chosen datasets, training with SGD for just a single epoch is unnaturally challenging, as the authors themselves convincingly argue (e.g., on p7). Why did the authors choose to make such clearly unfair comparisons? Even if a careful reader might be able to find the subtle details somewhere in the paper, it is safe to assume many readers will miss this (e.g., as I myself did).
> > > >
> > > > I wonder why the authors considered the “single epoch” setting at all. The proposed ModelZoo approach assumes that all data can be stored, but this is an assumption that does not seem to make any sense in the “single epoch” setting. The motivation behind this setting is that each sample can only be visited once, but this is clearly violated by storing all data and later revisiting it. It seems to me that by making this illogical assumption in the “single epoch” setting, misleadingly large gains over existing methods can be obtained. (Also, none of the compared methods store all of the data, which by itself is another reason that these comparisons are questionable.)
> > > >
> > > > I’m afraid there are more issues with the comparisons reported in the paper:
> > > >
> > > > -	The authors repeatedly argue that because in ‘ModelZoo-small’ they use “an extremely small” network, that any performance increase they find “surely” cannot be from the architecture. As I pointed out before, this is just not true, because larger architectures are not necessarily better. From Figure 1, this indeed seems to be the case: compare the black line after the first task with the faint lines after the first task.
> > > > -	Even though the authors strongly emphasize how small the networks used in ‘ModelZoo-small’ are, it actually seems to be the case that the total number of parameters used by ‘ModelZoo-small’ is still >50% higher than that of the methods compared against (see Table 2).
> > > > -	Finally, I repeat the concern from my initial review that some results are simply taken from other publications, which likely used different hyperparameter-settings / architectures / training protocols. How can I know that these comparisons are fair or reasonable? Moreover, it still appears to be the case that it is not correctly reported in the paper where all results come from. (For example, many of the reported results in Table 1 do not have any "stars" associated with them to indicate they were taken from other publications, but from the authors’ rebuttal I understand that they were taken from elsewhere?) I would not be able to trace back the origin of the reported results.
> > > > (Another note is that, as a reviewer for this paper, I should not be required to have to track down other papers in order to establish whether reported comparisons are fair or not. All relevant details should be reported in the paper itself.)

---

> > > > > ### Author Response · Authors · 2021-11-23
> > > > > **Response (1/3)**
> > > > >
> > > > > **>> I had indeed misinterpreted some of the reported experiments, my apologies.**
> > > > >
> > > > > Glad to clarify.
> > > > >
> > > > > But we believe there are still a number of misunderstandings in how the reviewer is interpreting our results. We evaluate Model Zoo for many different settings in this paper (single epoch, multi-epoch, no replay, 10% replay, small base model, large base model, identical base model as competing methods etc.). Different competing methods in the literature use different evaluation sttings. **We show that for each of the considered competing methods, Model Zoo outperforms that method when evaluated using the particular method’s problem setting.  All comparisons in the paper are therefore totally fair.** **We request the reviewer to bear in mind this gamut of problem settings and to not compare, for example, Setting 1 of Model Zoo versus Setting 2 of a competing method--- the comparison will of course be unfair that way.**
> > > > >
> > > > > Finally, we appreciate the critical eye of the reviewer but we would encourage them to zoom out and appraise these results. **Our paper presents a sound method that works across 7 datasets, 3 architectures, more than 6 different problem settings, and compares very favorably to existing methods across different problem settings---all of these with a single set of hyper-parameters.**
> > > > >
> > > > > We next clarify some of the points raised by the reviewer.
> > > > >
> > > > >
> > > > > **>>> The authors repeatedly argue that because in ‘ModelZoo-small’ they use “an extremely small” network, that any performance increase they find “surely” cannot be from the architecture. As I pointed out before, this is just not true, because larger architectures are not necessarily better. From Figure 1, this indeed seems to be the case: compare the black line after the first task with the faint lines after the first task.**
> > > > >
> > > > > The black line in Fig 1a (Isolated-small-single-epoch) learns the first task better than the faint lines (which have larger models). This may suggest that small architectures can be “better” but it is not true---it is an artifact of the single epoch setting because the small architecture (Isolated) is relatively less undertrained. **This is not true for the multi-epoch setting---larger model if trained well is indeed better in this case.** It is not meaningful to argue whether a large or small model is better if none of them are trained to convergence.
> > > > >
> > > > > **Please see our previous response for additional experiments using the Resnet-18-S architecture used in some competing methods.** We show that we can preserve the substantial accuracy gains of Model Zoo using WRN-16-4 and Small architectures and using the new architecture Resnet-18-S as well. For the multi-epoch setting, Model Zoo and Isolated are better than competing methods. It is important to note that we achieve similar accuracy in the multi-epoch setting for both WRN-16-4 and Resnet-18-S; this indicates that the performance improvement is not driven by architecture but primarily by our method.
> > > > >
> > > > > Also see the other argument (“Another evidence as seen from Table 2 of the manuscript”).
> > > > >
> > > > > **>> this is just not true, because larger architectures are not necessarily better. From Figure 1, this indeed seems to be the case: compare the black line after the first task with the faint lines after the first task.**
> > > > >
> > > > > We disagree. Larger architectures---**if they are well trained well**---have to perform at least as well as a small architecture. Mathematically, the hypothesis class of a smaller network is a subset of the hypothesis class of the larger network, so a larger network should be better if it is well-trained, e.g., the double descent phenomenon suggests that over-parameterized deep networks generalize better. This is also seen in Table 1 and Fig 1a: Model Zoo with a larger network is better than its smaller variant---if both are trained for multiple epochs. Our point in Fig 1a is that methods that train for one epoch do not end up training their models well, small or large.
> > > > >
> > > > > **>>> Even though the authors strongly emphasize how small the networks used in ‘ModelZoo-small’ are, it actually seems to be the case that the total number of parameters used by ‘ModelZoo-small’ is still >50% higher than that of the methods compared against (see Table 2).**
> > > > >
> > > > > Yes, this is a minor increase in the number of parameters that comes with a more than 10% boost in accuracy, even in the single epoch setting. It has been reported before that these methods have *worse accuracy* if trained for multiple epochs (A-GEM, Appendix F https://arxiv.org/abs/1812.00420); in comparison Model Zoo gets about 20% boost in accuracy over its single epoch variant. Even compared to RMN which has 5x more weights, Model Zoo/Isolated-small’s accuracy is larger by more than 10%. This suggests that Model Zoo is an effective continual learner, no matter how we parse its accuracy.

---

> > > > > > ### Comment · Reviewer_R6VS · 2021-11-23
> > > > > > **Clarification of my concerns about the comparisons**
> > > > > >
> > > > > > Thank you for your quick response.
> > > > > >
> > > > > > ***--> Finally, we appreciate the critical eye of the reviewer but we would encourage them to zoom out and appraise these results.***
> > > > > >
> > > > > > I want to ensure the authors that I do appreciate the larger picture of this work. As indicated in my original review, I think this paper provides important insights and food-for-thought. I certainly think this work is interesting/important enough for publication at this conference. If this was not the case, I would not be spending so much effort. However, to be published at this conference, the paper also needs to be correct, it should not make unsupported claims and it should not contain unfair or misleading comparisons.
> > > > > >
> > > > > > Let’s look at some concrete examples.
> > > > > >
> > > > > > In Table 1, to which of your variants could the results of the method “ER-Reservoir” be compared in a fair manner?
> > > > > >
> > > > > > In Figure 1, the caption is “How well do existing continual learning methods work?”. But the only existing continual learning methods that are included are trained for just a single epoch. It is clear that continual learning methods that are trained for multiple epochs could do a lot better. This is a more general trend in the paper. When the authors talk about “existing continual learning methods”, this essentially refers to methods trained in the single epoch setting.
> > > > > >
> > > > > >
> > > > > > ***--> This may suggest that small architectures can be “better” but it is not true---it is an artifact of the single epoch setting because the small architecture (Isolated) is relatively less undertrained.***
> > > > > >
> > > > > > I agree that potential benefits of small networks are likely more pronounced in the single epoch setting. But this only strengthens my concern, given that a large proportion of the comparisons in this paper are in this single epoch setting.
> > > > > >
> > > > > >
> > > > > > ***--> We hope that the reviewer appreciates that the continual learning literature is not yet streamlined enough for a reader to be able to read one paper without reading other papers preceding it.***
> > > > > >
> > > > > > I disagree with this. I think it is important for any published paper to be as self-contained as possible.
> > > > > >
> > > > > >
> > > > > >
> > > > > > ***--> The point here was to show how much performance is left on the table by (i) not splitting the capacity, (ii) not reusing past data, and (iii) under-training models.***
> > > > > >
> > > > > > Thanks. This is well-formulated. I hope this can help make my underlying concerns clear. I think it is trivial and/or well-established in the literature that performance of continual learning methods can be substantially increased by (i) reusing past data and by (ii) allowing training for multiple epochs. To me, the (main) contribution of this paper is about how splitting the capacity of an algorithm can be helpful. However, for most of the reported comparisons, the effect of splitting capacity is confounded by the effects of reusing past data and/or training for multiple epochs.

---

> > > > > > > ### Author Response · Authors · 2021-11-23
> > > > > > > **Response (1/2)**
> > > > > > >
> > > > > > >
> > > > > > > We appreciate your time and effort. But we are still worried that there are a number of misunderstandings/misreadings that is causing you to believe that there are “unsupported claims or that our paper contains unfair or misleading comparisons“.
> > > > > > >
> > > > > > > **>> When the authors talk about “existing continual learning methods”, this essentially refers to methods trained in the single epoch setting.**
> > > > > > > **>> a large proportion of the comparisons in this paper are in this single epoch setting.**
> > > > > > >
> > > > > > > **This is not true.** There are a number of existing methods that train for multiple epochs throughout the paper (FRCL, FROMP, RMN, APD to name a few). **Please look at Table A1. We have compared to over 13 methods that evaluate in the multi-epoch setting.**
> > > > > > >
> > > > > > > **>> In Table 1, to which of your variants could the results of the method “ER-Reservoir” be compared in a fair manner?**
> > > > > > >
> > > > > > > As the reviewer themselves have pointed out before “replaying data and yet training for a single/few epochs” does not really make sense. Note that we have marked ER-reservoir with the $\dagger$ symbol in Table 1 to indicate that their training is done for 1-5 epochs (they show both settings in their paper).
> > > > > > >
> > > > > > > Here are three points of comparison that our paper provides to ER-Reservoir.
> > > > > > >
> > > > > > > 1. **Single epoch setting:** Isolated-small (single epoch) in Fig 1a obtains 66% on Split-MiniImagenet (even without any data replay, in the single epoch setting) as opposed to ER-Reservoir which gets 64.03% with data replay (see Table 1). Our method has a large disadvantage (no data replay) and yet it outperforms ER-Reservoir.
> > > > > > > 2. **Multiple Epochs, No replay:** For Isolated-small (no data storage at all, multiple epochs), we get 90.18% on Split-CIFAR100 whereas ER-Reservoir gets 68.5%. This setting is conceptually fair because ER-Reservoir replays past data multiple times~~.~~
> > > > > > > 3. **Multiple Epochs, 10% replay:** Model Zoo-small (10% replay) gets 89.76% on Split-CIFAR100 while ER-Reservoir gets 68.5% with the same amount of replay. This is also fair because Model Zoo and ER-Reservoir have access to the same amount of past data and both are allowed to use the past data multiple times. Model Zoo simply also uses the current data multiple times.
> > > > > > >
> > > > > > >
> > > > > > > **>> It is clear that continual learning methods that are trained for multiple epochs could do a lot better. This is a more general trend in the paper.**
> > > > > > >
> > > > > > > **This is a contribution of our paper. It was not “clear” in the literature before our paper.** Not all methods do better in continual learning simply by training for multiple epochs. As pointed out in Appendix F (A-GEM https://arxiv.org/abs/1812.00420), this causes forgetting on the past tasks and therefore a lot of existing methods do not train for multiple epochs. It is a merit of our methods that we do not suffer from forgetting.
> > > > > > >
> > > > > > > **>> In Figure 1 ... the only existing continual learning methods that are included are trained for just a single epoch.**
> > > > > > >
> > > > > > > Yes, Figure 1 compares to continual learning methods in the single epoch setting as mentioned in the caption. As the caption says, the point of Fig 1a is not to claim “our method is better than other methods” but instead show that training models for a single epoch is not sufficient. We believe you are misreading the caption. If you’d like, we can remove the phrase “How well do existing continual learning methods work?” from the caption.
> > > > > > >
> > > > > > > **We have many comparisons to methods in the multi-epoch setting. Table 1 and Table A1 show over 13 methods e.g. RMN, APD, MEGA-II that are trained for multiple epochs.**

---

> > > > > > > > ### Comment · Reviewer_R6VS · 2021-11-28
> > > > > > > > **Further clarification about my main concern**
> > > > > > > >
> > > > > > > > It seems that the main point the authors try to make in their rebuttals is that their paper contains interesting / reasonable comparisons. But I want to stress again that this is not my main concern. My main concern is that the paper (also) contains comparisons that are unfair or misleading.
> > > > > > > >
> > > > > > > > I still don’t understand why, in Table 1, the authors choose to compare their methods, which are trained for multiple epochs per task (at least all the variants in this table), against methods that are only trained for a single epoch per task?
> > > > > > > >
> > > > > > > > Related to this, I wanted to respond to one comment from the last author rebuttal.
> > > > > > > >
> > > > > > > > ***-> This*** *(i.e., demonstrating that training for multiple epochs can improve performance)* ***is a contribution of our paper. It was not “clear” in the literature before our paper. Not all methods do better in continual learning simply by training for multiple epochs. As pointed out in Appendix F (A-GEM https://arxiv.org/abs/1812.00420), this causes forgetting on the past tasks and therefore a lot of existing methods do not train for multiple epochs. It is a merit of our methods that we do not suffer from forgetting.***
> > > > > > > >
> > > > > > > > I strongly disagree with this. By imposing the restriction that each training example can only be seen once, the problem indisputably becomes harder. Perhaps some methods that are optimized for the single epoch will perform worse when trained for multiple epochs, but methods optimized for multiple epochs should certainly perform better than (or at the very worst, comparable to) their counterpart methods in the single epoch setting.

---

> > > > > > > > > ### Author Response · Authors · 2021-11-28
> > > > > > > > > **Response**
> > > > > > > > >
> > > > > > > > > **>> My main concern is that the paper (also) contains comparisons that are unfair or misleading. I still don’t understand why, in Table 1, the authors choose to compare their methods, which are trained for multiple epochs per task (at least all the variants in this table), against methods that are only trained for a single epoch per task?**
> > > > > > > > >
> > > > > > > > > **Table 1 compares our method against a number of existing methods (both single epoch and multiple epoch)**. The caption makes it extremely clear how the other methods are structured and what they should be compared to. Such tables with captions that explain the different settings have been done in other continual learning papers before, e.g., https://arxiv.org/abs/2102.11343. In principle, we agree that such tables could be stratified to reflect different settings. But it is very hard to do so because the current literature is too fragmented and there are just too many different settings.
> > > > > > > > >
> > > > > > > > > **Our paper does not contain unfair comparisons. Tables 1 and A1 contain MANY comparisons. This is in the interest of completeness. An informed reader who understands what different settings mean can read the row of a competing method with the corresponding row in the table for Model Zoo (the caption explains this).** **One cannot read an arbitrary pair of rows in Table 1 and say that the comparison is unfair; this would be like comparing apples to oranges. Please read the caption.**
> > > > > > > > >
> > > > > > > > > **There is more to these tables than simply saying “one method works better than all methods”.** In fact this is not what we are saying, we would not have shown multiple rows for Model Zoo variants in that case. Different existing methods use different problem settings. That is why Fig 1 and Table 1 show the performnce of our method under different settings and competing methods of different settings.
> > > > > > > > >
> > > > > > > > > We repeat this paragraph from the previous response. Existing methods operate under different settings and for sake of completeness we have included them all. In particular to address your point about single epoch methods, **Table 1 and Table A1 also contain methods that are trained for multiple epochs** **(FRCL, FROMP, RMN, APD to name a few). In total, we have compared to 13 methods in the multi-epoch setting.**
> > > > > > > > >
> > > > > > > > > **We will add more rows with Model Zoo’s single epoch accuracy to Table 1 (same data as that of Fig 1a) in the final version of the paper. Perhaps this will make it even more clear that there are different points of comparison in the table.**
> > > > > > > > >
> > > > > > > > > **>>> I strongly disagree with this. By imposing the restriction that each training example can only be seen once, the problem indisputably becomes harder. Perhaps some methods that are optimized for the single epoch will perform worse when trained for multiple epochs, but methods optimized for multiple epochs should certainly perform better than (or at the very worst, comparable to) their counterpart methods in the single epoch setting.**
> > > > > > > > >
> > > > > > > > > We do not see the disagreement with the reviewer here. We agree with them. Our response said “Not all methods do better in continual learning simply by training for multiple epochs”. This is also what the reviewer is saying here.
> > > > > > > > >
> > > > > > > > > As Fig 1a and Table 1 show, it is a merit of our methods that they can be executed in both the single epoch setting as well as the multi epoch setting. Our method outperforms existing ones in each of these settings (seeing a datum only once = Isolated single epoch, multiple epochs = other variants).

---

> > > > > > > ### Author Response · Authors · 2021-11-23
> > > > > > > **Response (2/2)**
> > > > > > >
> > > > > > > **>> I agree that potential benefits of small networks are likely more pronounced in the single epoch setting. But this only strengthens my concern, given that a large proportion of the comparisons in this paper are in this single epoch setting.**
> > > > > > >
> > > > > > > This is some major misunderstanding here. Yes, small networks work work well in our methods. But large networks also work well in our methods. A large number of current methods cannot train models sufficiently lest they suffer from forgetting. We show that
> > > > > > >
> > > > > > > 1. if we train a small network for one epoch, our developed methods work better than existing methods;
> > > > > > > 2. if we train the same small network for multiple epochs, our developed methods work substantially better than existing methods;
> > > > > > > 3. if we train a large network for multiple epochs, our developed methods work substantially better than existing methods that also evaluate for multiple epochs.
> > > > > > >
> > > > > > > **Therefore, this is a completely exhaustive evaluation. It is not as the benefits of Model Zoo are restricted to small networks. It works in all settings that the literature has tackled before. In fact, as our results show, it works better for large networks trained for multiple epochs.**
> > > > > > >
> > > > > > > **Can we please request you to precisely formulate and state as to what your exact concern about the network size is?**
> > > > > > >
> > > > > > >
> > > > > > > **>> I think it is important for any published paper to be as self-contained as possible.**
> > > > > > >
> > > > > > > We agree with the reviewer here. Our paper has a lot of details; it is 28 pages right now including the Appendix. But please understand that we cannot possibly summarize the experimentation setup of such a large number of diverse papers in our manuscript. The continual learning literature is simply not enough streamlined yet to be able to even put all of the existing methods neatly into clear buckets. The best place to study a method is the paper of the original authors. **Our work is a step towards making this literature more systematic. Rather than picking one specific formulation, we have studied continual learning from a lot of different angles and we hope in the future more clarity on problem formulations will emerge in the literature.**
> > > > > > >
> > > > > > > **>> However, for most of the reported comparisons, the effect of splitting capacity is confounded by the effects of reusing past data and/or training for multiple epochs.**
> > > > > > >
> > > > > > > **This is not true.** If you want to ablate on “splitting capacity”, you can look at Model Zoo versus existing methods that train one network. If you want to ablate on “reusing past data”, you should look at Model Zoo with 10% data replay, or Isolated without any data replay. If you want to ablate on “training for multiple epochs“ you should look at the single epoch setting in Fig 1a, Table 2.
> > > > > > >
> > > > > > > This paper is a careful and systematic study of an effective strategy (that of splitting the capacity) that is motivated by theory. The experiments are thorough and there are extensive and fair comparisons to existing methods. **We believe the reviewer is not comparing apples to apples when they read our plots and tables. We understand that doing so is not easy because there is such a large amount of diversity in the existing methods. But this is not our fault. Our comparisons are exhaustive enough to cover most settings in the literature. We would just request the reviewer to carefully interpret them.**

---

> > > > > ### Author Response · Authors · 2021-11-23
> > > > > **Resopnse (2/3)**
> > > > >
> > > > > **>>> training with SGD for just a single epoch is unnaturally challenging, as the authors themselves convincingly argue (e.g., on p7).**
> > > > >
> > > > > We couldn’t agree more. This is exactly what we have argued in the paper and the rebuttal.
> > > > >
> > > > > **>>> Between the main results presented in Fig 1 and Table 1, most (or all, in case of Fig 1) of the compared methods are trained with a single epoch, while most (or all, in case of Table 1) of the Isolated/ModelZoo are trained for multiple epochs. It is clear that this is not a fair comparison: on the chosen datasets... Why did the authors choose to make such clearly unfair comparisons?**
> > > > >
> > > > > **For Isolated (single epoch), no data is replayed, so this is a completely apples-to-apples comparison.** In the context of the single epoch setting, Fig 1a, Table 1 and Table 2, say the following.
> > > > >
> > > > > 1. **Isolated (small CNN, black) for Split-MiniImagenet (Fig 1a) outperforms other methods even if it is trained for a single epoch.**
> > > > > 2. Isolated (WRN-16-4) for Split-MiniImagenet (Fig 1a) is worse than other methods if trained for a single epoch.
> > > > >     1. If the same architecture is trained for multiple epochs (note: no data replay here), it is better than all methods. Other methods could have also trained for multiple epochs but as Appendix F in the A-GEM paper (https://arxiv.org/abs/1812.00420) reports, they suffer a large drop in accuracy if they do so.
> > > > >     2. If the same architecture is trained on a different problem (Fig A9 in the Appendix for Coarse-CIFAR100 and Split-CIFAR100), Isolated-small (single epoch, i.e., no data replay whatsoever) handily outperforms all existing methods. Single epoch models are severely undertrained and this is the reason for different trends for different datasets. **This discrepancy disappears if models are trained till convergence, e.g., in Fig 1 and Fig A9, multi-epoch Isolated outperforms existing methods.** Again note that none of the competing single-epoch methods (e.g., GEM, A-GEM, see more in Table A1) have been demonstrated to work well if they are trained for multiple epochs on each task.
> > > > > 3. A-GEM in Fig 1a uses 10% of the past data. So the appropriate comparison here is Model Zoo-small (10% replay) in Table 1, which handily outperforms A-GEM.
> > > > > 4. In addition to using data only once, another motivation for training for a single epoch in the existing literature is to restrict the training time of the continual learner. This is the reason we also ran Model Zoo in the setting where each new task is trained for one epoch only but past data is still replayed. As Table 2 shows, Model Zoo-small still has a better training time than say, A-GEM.
> > > > >
> > > > > **Therefore, our comparisons are not unfair at all. We just have a number of different comparisons to address the diverse evaluation settings in the existing literature. One must compare the correct variant of our method depending upon which competing method is being talked about; not all competing methods are single-epoch and some of them also use past data. Please also review the caption of Fig 1a: the purpose of this figure is not to simply say “our method is better than other methods”.**
> > > > >
> > > > > We appreciate that the reviewer is trying to study these results carefully. But as they can see, the diversity of settings in the literature is really immense. This is exactly the reason why simple methods like Model Zoo, or even Isolated, are useful for the field to progress. And this is why we believe our work deserves to be published. Instead of using one particular evaluation setting and comparing against only the corresponding methods, we study the problem from different angles in this paper. This makes reading the paper more complicated, yes, but it is also a more useful contribution to the literature this way.
> > > > >
> > > > > **>> Also, none of the compared methods store all of the data, which by itself is another reason that these comparisons are questionable.**
> > > > >
> > > > > The Isolated learner does not store any data either. And yet it outperforms existing methods. The point here was to show how much performance is left on the table by (i) not splitting the capacity, (ii) not reusing past data, and (iii) under-training models. A-GEM stores 10% of the past data and we see in Table 1 that Model Zoo compares favorably to it.
> > > > >
> > > > > **>>> Even if a careful reader might be able to find the subtle details somewhere in the paper, it is safe to assume many readers will miss this**
> > > > >
> > > > > We have made all distinctions explicit at multiple places in the paper. Entries in Table 1 and legend of Fig 1 or Fig B.7 use markers to denote that the methods were evaluated in the single-epoch setting and the caption mentions the same. Table 2 and Section 4.3 (iii) also clearly mention the single epoch setting. We believe that a careful reader will be able to appreciate the details.

---

> > > > > ### Author Response · Authors · 2021-11-23
> > > > > **Response (3/3)**
> > > > >
> > > > > **>>> Moreover, it still appears to be the case that it is not correctly reported in the paper where all results come from. (For example, many of the reported results in Table 1 do not have any "stars" associated with them to indicate they were taken from other publications, but from the authors’ rebuttal I understand that they were taken from elsewhere?) I would not be able to trace back the origin of the reported results.**
> > > > >
> > > > > **We believe the reviewer misunderstood the last sentence of the caption. We have updated the caption to make it even more clear.**
> > > > >
> > > > > 1. **All numbers without any markers are taken from the papers cited in the first column.**
> > > > > 2. Numbers for other methods on Split-MiniImagenet were computed by us using open-source implementations of the original authors. This is because the original authors did not evaluate on these datasets.
> > > > > 3. As the caption says, the star (*) marker denotes that these numbers are not from the original paper (first column) but instead from one of the three papers Nguyen et al., 2017; Serra et al., 2018; Chaudhry et al., 2019a.
> > > > > 4. A lot of existing papers do not report continual learning curves. So we ran the code of the original authors to obtain the curves Fig 1 and Fig A9. This is the same code as point 2 above.
> > > > >
> > > > >
> > > > > **>>> (Another note is that, as a reviewer for this paper, I should not be required to have to track down other papers in order to establish whether reported comparisons are fair or not. All relevant details should be reported in the paper itself.)**
> > > > >
> > > > > We appreciate this point. We have mentioned all the settings used by our work clearly in the paper. The number of existing methods and the diversity of their settings is vast, the original papers that we have cited extensively do an excellent job of clarifying the problem setting. We hope that the reviewer appreciates that the continual learning literature is not yet streamlined enough for a reader to be able to read one paper without reading other papers preceding it.
> > > > >
> > > > > **>>> Finally, I repeat the concern from my initial review that some results are simply taken from other publications, which likely used different hyperparameter-settings / architectures / training protocols.** **How can I know that these comparisons are fair or reasonable?**
> > > > >
> > > > > There are a lot of existing methods, a lot of existing networks, and training methods. As we explained, we have reported extensive comparisons with existing architectures and existing problem settings. Our training methodology does not do anything special (Appendix A.3). All hyper-parameters are fixed for all experiments. We cannot run the competing methods in settings that they were not designed to be run in---that would be unfair comparison. **Does the reviewer have any specific suspicion about our hyper-parameter settings or training protocols?**
> > > > >
> > > > > **Our paper presents ample evidence for a sound method that works across 7 datasets, 3 architectures, more than 6 different problem settings, and compares very favorably to existing methods across these different problem settings---all of these with a single set of hyper-parameters.**

---

> > > ### Author Response · Authors · 2021-11-20
> > > **Response (2/2)**
> > >
> > > **>>> But I think the problem is more severe than this, because based on Figure 1 (Left), it even seems likely that a large part of the reported improvements are due to these differences in architecture!**
> > >
> > > We next address your concerns regarding Fig 1a, and explain why this does not contradict the observations made from the two tables above. We believe there is some misunderstanding in how the reviewer is reading this plot.
> > >
> > > **>>> Strikingly, after the first task, the performance of the Model Zoo methods (~90% for large and ~60% for small) is substantially higher than the performance of the existing methods that are compared against (35-50%).  These differences after the first task seem to explain most, if not all, of the reported performance improvements after all tasks have been learned.**
> > >
> > > **After the first task, Model Zoo (single-epoch WRN-16-4 in cyan) is at 25%, not 90%.** **This is worse than competing methods after the first task**. The striking thing is that even though Model Zoo (single epoch WRN-16-4) is worse than all other methods at the start, **the same model is better than other methods at the end of continual learning.** This indicates that the performance improvement is a result of being an effective continual learner, not just architecture.
> > >
> > > The 90% number that the reviewer is pointing to is for Model Zoo/Isolated trained for multiple epochs (lines in red/orange). As the caption says, the multi-epoch curves were included to highlight how under-trained all methods are in the single-epoch setting. **Current methods that train models for a single epoch do not end up training the networks to convergence.** In fact, as reported in the A-GEM paper (Appendix F, https://arxiv.org/abs/1812.00420) these methods suffer a degradation of accuracy if models are trained for multiple epochs.
> > >
> > > **>>> Let’s look at the performance of all compared methods after just the first task...**
> > >
> > > If all models are trained for one epoch only, after the first task, Isolated/Model Zoo (WRN-16-4) has ~25% accuracy while competing methods have a higher 35-50% accuracy.
> > >
> > > But the exact same network Isolated/Model Zoo (WRN-16-4) if trained to convergence for multiple epochs has ~90% accuracy after the first task. Isolated/Model Zoo (small) has much fewer weights than any of these models. If trained for a single epoch, its initial accuracy is ~60% (black/dark blue line in Fig 1a). This makes sense because a smaller model is relatively better-trained after 1 epoch. We have updated the legend of Fig 1a to clarify that black/dark blue lines are also for the single epoch setting.
> > >
> > > **>>> I’m afraid that it is possible – and probably likely – that most of the reported performance improvements in this paper are due to differences in network architecture (or perhaps optimization) that are arguably not interesting from a continual learning perspective.**
> > >
> > > We hope our response eliminates this concern. All evidence points to the performance coming not from the architecture but instead from continual learning.
> > >
> > > The confusion in the reviewer’s mind perhaps originates from the fact that competing methods in Fig 1a train for a single epoch at each episode (this naturally alleviates forgetting, see Appendix F in AGEM https://arxiv.org/abs/1812.00420) while we train for both single/multiple epochs at each episode. As we explained, not only does our method outperform existing ones in the single-epoch setting, but more importantly, current methods in their effort to reduce forgetting lead to severely undertrained models and leave a lot of performance on the table. Model Zoo does not suffer from forgetting even if it is trained for multiple epochs.

---

> > ### Comment · Reviewer_R6VS · 2021-11-19
> > **Comment on the title**
> >
> > Thank you for clarifying your motivation for the use of "a growing brain" in the title. I accept that when clearly used in a colloquial manner, it is acceptable to refer to the proposed Model Zoo algorithm as a growing "brain". However, as presented currently, I don't think it is clear from the title that "brain" is used colloquially. It would be acceptable to me if the term brain is put in quotation marks in the title.
> >
> > I appreciate that in the machine learning literature examples can be found that make similar comparisons to the brain without proper justification, but I don't think that is a valid argument to do the same here.

---

> > > ### Author Response · Authors · 2021-11-20
> > > **Re: Comment on the title**
> > >
> > > **>>> It would be acceptable to me if the term brain is put in quotation marks in the title**
> > >
> > > This is reasonable. We have updated the manuscript with the word brain in quotation marks.

---

> ### Author Response · Authors · 2021-11-18
> **Response to Reviewer R6VS (2/3)**
>
> **>>>  I am particularly worried about this because from Fig 1A, it seems to be the case that the main part of the reported improvement over existing methods is due to an improved ability to learn individual task (e.g. compare all accuracies after just the first), rather than due to an improvement in forward/backward transfer.**
>
> We are a bit unsure of what you mean by “compare all accuracies after just the first” in this comment.
>
> If you refer to the flat-ish lines in Fig 1a, note that the Y-axis of Fig 1a is the average accuracy on *all the past tasks*. So if the curve is flat, it indicates that the average accuracy after the new task is fitted, is as high as the average accuracy without the current task. This flat curve can result from not just the ability to learn the new task very well but also come from backward transfer on past tasks, if the accuracy of the new task is less than the average accuracy of the past tasks. We have noticed that the curve drops a bit for harder problems, e.g., Coarse-CIFAR100 in Fig A9 (left) where the tasks are semantically different from each other but the curve is either flat or increasing for simpler problems like Split-CIFAR100 (Fig A9 right) where task are semantically heterogeneous. Even a marginally positive slope on this curve indicates that *all past tasks* improve their accuracy as more episodes are shown. The goal of Fig 1a was to show that (i) even Isolated-small is competitive with existing methods in terms of average per-task accuracy and (ii) that the single-epoch setting that is employed in a number of existing methods cannot sufficiently leverage data from the tasks because it creates undertrained models.
>
> One can study whether continual learning is happening or not using Fig 1b which shows the forward and backward transfer of 5 particular tasks in Split-MiniImagenet (see Fig. A8 for the extended version). Each of these tasks has a strong backward transfer because the accuracy of all curves increases with episodes. Fig. 1b also displays strong forward transfer because the cross marks (which denote the accuracy of an Isolated learner) are often below the continual learning curves. The forward-backward transfer is more significant for the single-epoch setting (cyan in Fig 1a and dotted lines in Fig 1b) but one must remember that any deep learning model is severely undertrained if training concludes after one epoch. The multi-epoch setting is more meaningful and even if the initial accuracy of the task is quite high in Fig 1b, there is strong forward and backward transfer going on (see the multi-epoch columns in Table 2).
>
> **>>> An important class of methods that is not considered in this paper is generative replay... I think it is important to atleast discuss these methods as I am positive that some more recent generative replay variants (e.g. van de Ven et al., 2020 https://www.nature.com/articles/s41467-020-17866-2; Liu et al., 2020 https://arxiv.org/abs/2004.09199) could outperform the Isolated baseline.**
>
> Thank you for pointing us to literature on generative replay. We have included these citations in the related work section of the updated manuscript. Both Van den Ven et al,. 2020 and Liu et al., 2020 focus on the class-incremental setting, i.e., where new classes are shown to the learner in successive episodes. Our paper instead focuses on the task incremental setting where each episode provides a new task (a set of classes).
>
> That said, there is one experiment in Van de Ven et al., 2020 on a slightly different problem, namely Split-CIFAR100 with 10 classes per task (Fig 8b). It is difficult to obtain the exact numbers because they only present it as a bar plot but the methods proposed in their paper do not achieve more than 80% accuracy on this problem. All numbers reported in our Table 1 (including those for Isolated) are about 89% for Splitt-CIFAR100. Note that these numbers are not directly comparable, our setting (which a lot of other papers also employ) has 20 tasks of 5 classes each. Their setting has 10 tasks of 10 classes each. The former has easier tasks but more episodes, the latter has harder tasks but fewer episodes.
>
> Two of your suggested references (https://arxiv.org/abs/1904.07734 and https://www.nature.com/articles/s41467-020-17866-2.pdf) argue for data replay in continual learning; thanks for these references which are cited in our paper now.

---

> ### Author Response · Authors · 2021-11-18
> **Response to Reviewer R6VS (3/3)**
>
> **>>>  It is claimed that the “Multi-Head (multi-task)” baseline is not a continual learner because it is jointly trained on all tasks.  However, it seems straight-forward to adapt this baseline in such a way that it is, for the purposes of this paper, a “legitimate continual learner”**
>
> We completely agree with the reviewer that multi-head can be converted into a continual learner. Our goal in using Multi-Head (multi-task) was to estimate some kind of an upper bound on the accuracy of Model Zoo. And the results of doing so are quite interesting. As Table 1 shows, for some problems (Split-CIFAR10 and Split-CIFAR100) such a multi-task learner is marginally better than Model Zoo but it performs worse on harder problems like Coarse-CIFAR100 and Split-MiniImagenet. This indicates that splitting the capacity in Model Zoo is effective.
>
> The task-incremental multi-task learner that the reviewer suggests is similar to the batch-size (b) in Model Zoo increasing with each episode. We can estimate the accuracy of such an algorithm using Fig 4 (middle). For b <= 10, the first 10 episodes use all the past tasks while the last 10 episodes use 10 tasks with the loss loss. The accuracy at the end of the continual learning problem (i.e., after 20 episodes for Split-CIFAR-100 and Split-MiniImagenet) is 74.13 and 84.9. Both these numbers are much worse than those in Table 1 for Model Zoo.
>
> The two paragraphs above together indicate that the continual Multi-Head learner is likely to perform well but not as well as Model Zoo. We believe that this strategy could be more useful for the class-incremental setting where we can design a class-specific boosting mechanism on the same model/set of models instead of adding a new model to the ensemble at each round of continual learning. We hope to take our future work in this direction, so thank you for your suggestions.
>
> **>>> It is unclear to me how Model Zoo with limited replay is performed, and it would be good to provide some more details. In the text it is stated that only a subset of the data is used in equation (7), but I think the authors probably meant to refer to another equation?**
>
> You are right, (7) only refers to calculating the accuracy of a particular task. In Section 4.2, we now point the reader to Appendix A.4 which discusses how experiments with limited replay are performed. We elaborate upon this next.
>
> When the task (say task A) is first seen, Model Zoo is allowed to use all available data. For all future episodes, if Model Zoo picks a past task to retrain with, such a retraining uses only a fixed subset of the tasks’ data (10% of the samples are selected at random for this purpose). We sample each mini-batch to contain an equal number of samples from all past and current task. At inference time, the member of Model Zoo that is trained on all data of task A (this is the model that was fitted when task A was first shown to the continual learner) is assigned a proportionately larger weight in (equation 7). For 10% replay, this will amount to 10x larger weight that other models which used 10% data from task A. Mathematically, both of these training and inference modifications are equivalent to using coefficients that scale up the loss of the past task depending upon the number of samples that it has.
>
> **>>> The claim in the title that the proposed Model Zoo is a “Growing Brain” does not appear to be supported or justified anywhere in the text. I think it would be good to either modify the title, or to justify and discuss the reasons for this comparison in the text.**
>
> We wanted to emphasize the fact that Model Zoo grows the size of the ensemble as it sees successive episodes and this increasing learning capacity (and thereby splitting the capacity of the larger model) is effective for continual learning. There are similar usages of the phrase in existing papers, e.g., https://arxiv.org/abs/1907.07844. We agree with you that this is a colloquial usage of the phrase “growing a brain” but if you will allow us to keep it as the title, we will be very glad. We have added a sentence at the end of Section 3 that discusses the title.

---

### Author Response · Authors · 2021-11-18
**Common response to all reviewers**

We thank the reviewers for their detailed, careful and elaborate feedback. We are glad that the reviewers found that our work presents important theoretical and empirical insights and asks critical questions (Reviewers R6VS, LD4g, Cn5V), and has strong results with a large amount of experiments (Reviewer LD4g).

The main concerns that were raised were clarification of the problem setting and more details on how comparisons against existing methods were conducted (Reviewer R6VS), improving the writing to better reflect the diversity and motivations of existing continual learning approaches (Reviewer KWuM), missing citations/comparisons to existing works and clarification of methodology (Reviewer Cn5V). In particular, we appreciate the feedback of Reviewer KWuM about improving the writing and framing to better reflect some nuances in the literature. We have modified the text in a number of places to remove adverbs and phrases and our goal is to provide due deference to existing work. All modifications are annotated in blue or using strikethrough.

We believe we have addressed all the questions raised by the reviewers and would encourage them to engage in a discussion if there are additional concerns. We look forward to the discussion and revised scores of the reviewers. We believe our paper presents a careful and systematic study of continual learning and these insights are crucial for consolidating/evaluating our progress in the field and will likely lead to systematic future work. We would encourage the reviewers to champion our paper.

---

### Decision · Program_Chairs · 2022-01-20

**Decision:**

Accept (Poster)

**Comment:**

The paper sparked a very substantial discussion, not just about its scientific content, but also, as the authors will have seen, from the narrative standpoint. I would like to thank the reviewers who devoted time and efforts to discuss the paper’s content. I encourage the authors to polish further their paper prior to camera ready, using the numerous scientific comments made (e.g. R6VS, KWuM, the refs of Cn5V).

AC.